# Provable Maximum Entropy Manifold Exploration via Diffusion Models

**Riccardo De Santi** [* 1 2]  **Marin Vlastelica** [* 1 2]  **Ya-Ping Hsieh** [1]  **Zebang Shen** [1]  **Niao He** [1 2]  **Andreas Krause** [1 2]

## Abstract

Exploration is critical for solving real-world decision-making problems such as scientific discovery, where the objective is to generate truly novel designs rather than mimic existing data distributions. In this work, we address the challenge of leveraging the representational power of generative models for exploration without relying on explicit uncertainty quantification. We introduce a novel framework that casts exploration as entropy maximization over the approximate data manifold implicitly defined by a pre-trained diffusion model. Then, we present a novel principle for exploration based on density estimation, a problem well-known to be challenging in practice. To overcome this issue and render this method truly scalable, we leverage a fundamental connection between the entropy of the density induced by a diffusion model and its score function. Building on this, we develop an algorithm based on mirror descent that solves the exploration problem as sequential fine-tuning of a pre-trained diffusion model. We prove its convergence to the optimal exploratory diffusion model under realistic assumptions by leveraging recent understanding of mirror flows. Finally, we empirically evaluate our approach on both synthetic and high-dimensional text-to-image diffusion, demonstrating promising results.

## 1. Introduction

Recent progress in generative modeling, particularly the emergence of diffusion models (Sohl-Dickstein et al., 2015; Song & Ermon, 2019; Ho et al., 2020), has achieved unprecedented success in generating high-quality samples across diverse domains, including chemistry (Hoogeboom et al., 2022), biology (Corso et al., 2022), and robotics (Chi

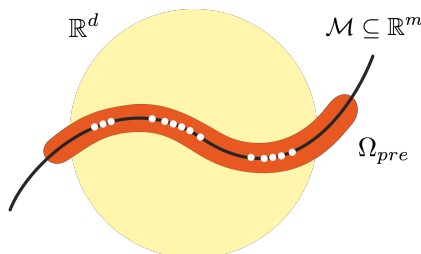

Figure 1: A diffusion model $\pi^{pre}$ pre-trained on a set of points (white) implicitly learns a set $\Omega_{pre}$ (orange) approximating the true low-dimensional data manifold $\mathcal{M} \subseteq \mathbb{R}^m$ (black) with $m \ll d$. The approximate data manifold $\Omega_{pre}$ can be significantly smaller than $\mathbb{R}^d$ (yellow).

et al., 2023). Traditionally, generative models have been employed to capture the underlying data distribution in high-dimensional spaces, facilitating processes such as molecule generation or material synthesis (Bilodeau et al., 2022; Zeni et al., 2023). However, simply approximating the data distribution is insufficient for real-world discovery, where exploration beyond high (data) density regions is essential.

Nonetheless, as illustrated in Figure 1, these models excel at capturing complex data manifolds that are often significantly lower-dimensional than the ambient space (Stanczuk et al., 2024; Kamkari et al., 2024; Chen et al., 2023), and can synthesize realistic novel samples that satisfy intricate constraints (e.g., valid drug molecules or materials). Yet, when the goal shifts to exploring novel regions within that manifold, a fundamental question remains:

*How can we leverage the representational power of generative models to guide exploration?*

**Our approach**  In this work, we tackle this challenge by first introducing the *maximum entropy manifold exploration* problem (Section 3). This involves learning a continuous-time reinforcement learning policy (Doya, 2000; Zhao et al., 2024) that governs a new diffusion model to optimally explore the approximate data manifold implicitly captured by a pre-trained model. To this end, we present a theoretically grounded algorithmic principle that enables self-guided exploration via a diffusion model's own representational power of the density it induces (Section 4). This turns exploration

---

[*]Equal contribution  [1]ETH Zurich, 8092 Zurich, Switzerland [2]ETH AI Center, Zurich, Switzerland. Correspondence to: Riccardo De Santi <rdesanti@ethz.ch>.

*Proceedings of the 42$^{nd}$ International Conference on Machine Learning*, Vancouver, Canada. PMLR 267, 2025. Copyright 2025 by the author(s).

into density estimation, a task well-known to be challenging in high-dimensional real-world settings (Song et al., 2020; Kingma et al., 2021; Skreta et al., 2024). To overcome this obstacle and render the method proposed truly scalable, we leverage a fundamental connection between the entropy of the density induced by a diffusion model and its score function. Building on this, we propose a practical algorithm that performs manifold exploration through sequential fine-tuning of the pre-trained model (Section 6). We provide theoretical convergence guarantees for optimal exploration in a simplified illustrative setting by interpreting the algorithm proposed as a mirror descent scheme (Nemirovskij & Yudin, 1983; Lu et al., 2018) (Section 5), and then generalize the analysis to realistic settings building on recent understanding of mirror flows (Hsieh et al., 2019) (Section 7). Finally, we provide an experimental evaluation of the proposed method, demonstrating its practical relevance on both synthetic and high-dimensional image data, where we leverage a pre-trained text-to-image diffusion model (Section 8).

**Our contributions** To sum up, in this work we present the following contributions:

- The maximum entropy manifold exploration problem, that captures the goal of exploration over the approximate data manifold implicitly represented by a pre-trained diffusion model (Section 3)
- A scalable algorithmic principle for manifold exploration that leverages the representational power of a pre-trained diffusion model (Section 4), and a theoretically grounded algorithm based on sequential fine-tuning (Section 6).
- Convergence guarantees for the algorithm presented both under simplified and realistic assumptions leveraging recent understanding of mirror flows (Sections 5 and 7).
- An experimental evaluation of the proposed method showcasing its practical relevance on both an illustrative task and a high-dimensional text-to-image setting (Section 8).

## 2. Background and Notation

**General Notation.** We denote with $\mathcal{X} \subseteq \mathbb{R}^d$ an arbitrary set. Then, we indicate the set of Borel probability measures on $\mathcal{X}$ with $\mathbb{P}(\mathcal{X})$, and the set of functionals over the set of probability measures $\mathbb{P}(\mathcal{X})$ as $\mathbb{F}(\mathcal{X})$. We write $d\mu = \rho dx$ to express that the density function of $\mu \in \mathbb{P}(\mathcal{X})$ with respect to the Lebesgue measure is $\rho$. Along this work, all integrals without an explicit measure are interpreted w.r.t. the Lebesgue measure. Given an integer $N$, we define $[N] \coloneqq \{1, \dots, N\}$. Moreover, for two densities $\mu, \nu \in \mathbb{P}(\mathcal{X})$, we denote with $D_{KL}(\mu, \nu)$ the forward Kullback–Leibler divergence between $\mu$ and $\nu$. Ultimately, we denote by $\mathcal{U}[0, a]$ the uniform density over the bounded set $[0, a]$ with $a \in \mathbb{R}_+$.

**Continuous-time diffusion models.** Diffusion models (DMs) are deep generative models that approximately sample a complex data distribution by learning from observations a dynamical system to map noise to novel valid data points (Song & Ermon, 2019). First, we introduce a forward stochastic differential equation (SDE) transforming to noise data points sampled from the data distribution $p_{data}$ :

$$\mathrm{d}X_t = f(X_t, t)\mathrm{d}t + g(t)\mathrm{d}B_t \text{ with } X_0 \sim p_{data} \quad (1)$$

where $X_t \in \mathbb{R}^d$ represents a $d$-dimensional point, $(B_t, t \geq 0)$ is $d$-dimensional Brownian motion, $f : \mathbb{R}_+ \times \mathbb{R}^d \to \mathbb{R}^d$ is a drift coefficient, and $g : \mathbb{R}_+ \to \mathbb{R}_+$ is a diffusion coefficient. We denote with $p_t$ the marginal density at time $t$. Given a time horizon $t > 0$, one can sample $X_t \sim p_t$ by running the forward SDE in Equation (1). We denote the time-reversal process by $X_t^{rev} \coloneqq X_{T-t}$ for $0 \leq t \leq T$, following the backward SDE:

$$\mathrm{d}X_t^{rev} = f^{rev}(X_t^{rev}, T - t)\mathrm{d}t + \eta g(T - t)\mathrm{d}B_t \quad (2)$$

with $f^{rev}(X_t^{rev}, T - t)$ corresponding to:

$$-f(X_t^{rev}, T - t) + \frac{1 + \eta^2}{2}g^2(T - t)\nabla_x \log p_{T-t}(X^{rev})$$

where $\nabla_x \log p_t(x)$ is the score function, and $\eta \in [0, 1]$. By following the backward SDE in Equation (2) from $X_T \sim p_T$, after $T$ steps one obtains $X_0 = X_T^{rev} \sim p_0 = p_{data}$. In practice, $p_T$ is replaced by a fixed and data-independent noise distribution $p_\infty \approx p_T$ for large $T$, typically a Gaussian[1], motivated by an asymptotic analysis of certain diffusion dynamics (Tang & Zhao, 2024).

**Score matching and generation.** Since the score function $\nabla_x \log p_t(x)$ is unknown, it is typically approximated by a neural network $s_\theta(x, t)$ learned by minimizing the MSE at points sampled according to the forward process, namely:

$$\mathcal{J}(\theta) \coloneqq \mathbb{E}_{t \sim \mathcal{U}[0,T]} \mathbb{E}_{x \sim p_t} \left[ \omega(t) \| s_\theta(x, t) - \nabla_x \log p_t(x) \|_2^2 \right]$$

where $\omega : [0, T] \to \mathbb{R}_{>0}$ is a weighting function. Crucially, this is equivalent to the denoising score matching objective (Vincent, 2011) consisting in estimating a minimizer $\theta^*$ of:

$$\mathbb{E}_{t \sim \mathcal{U}[0,T]} \omega(t) \mathbb{E}_{x_0 \sim p_0} \mathbb{E}_{x_t | x_0} \| s_\theta(x_t, t) - \nabla_{x_t} \log p_t(x_t | x_0) \|_2^2$$

where $p_t(\cdot \mid x_0)$ is the conditional distribution of $x_t$ given an initial sample $x_0 \sim p_0$, which has a closed-form for typical diffusion dynamics. Once an approximate score function $s_{\theta^*}$ is learned, it can generate novel points approximately sampled from the data distribution. This is achieved by sampling an initial noise sample $X_0^\leftarrow \sim p_\infty$ and following the backward SDE in Equation (2), replacing

---

[1] In the following, we will choose $p_\infty$ to be a truncated Gaussian for the sake of theoretical analysis.

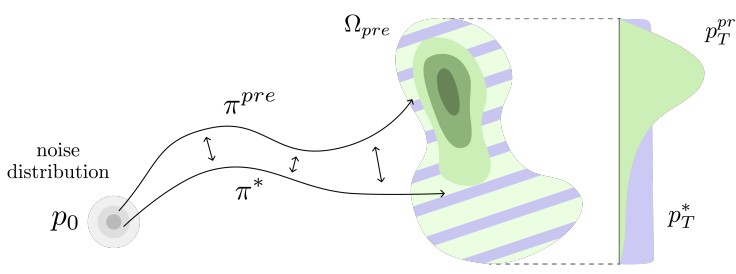
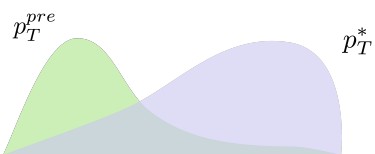

(a) Pre-trained and fine-tuned diffusion model processes.

(b) Regularized surprise maximization

Figure 2: (2a) Diffusion processes and marginal densities corresponding to the pre-trained model $\pi^{pre}$ (green), and maximally explorative fine-tuned model $\pi^*$ (violet). (2b) Fine-tuning a pre-trained diffusion model (green) via the surprise maximization principle in Eq. (9) one obtains a diffusion model (violet) able to sample low-density regions.

the true score $\nabla_x \log p_t(x)$ with $s_{\theta^*}$, leading to the process $\{X_t^{\leftarrow}\}_{t \in [0,T]}$. Next, we introduce a framework that we will leverage to fine-tune a pre-trained diffusion model.

**Continuous-time reinforcement learning.** We formulate finite-horizon continuous-time reinforcement learning (RL) as a specific class of stochastic control problems (Wang et al., 2020; Jia & Zhou, 2022; Zhao et al., 2024). Given a state space $\mathcal{X}$ and an action space $\mathcal{A}$, we consider the transition dynamics governed by the following diffusion process, where we invert the direction of the time variable:

$$d\overline{X}_t = b(\overline{X}_t, t, a_t)dt + \sigma(t)dB_t \text{ with } \overline{X}_0 \sim \mu \quad (3)$$

where $\mu \in \mathbb{P}(\mathcal{X})$ is an initial state distribution, $(B_t, t \geq 0)$ is $d$-dimensional Brownian motion, $b : \mathcal{X} \times \mathcal{A} \to \mathbb{R}^d$ is the drift coefficient, $\sigma : [0, T] \to \mathbb{R}_+$ is the diffusion coefficient, and $a_t \in \mathcal{A}$ is a selected action. In the following, we consider a state space $\mathcal{X} := \mathbb{R}^d \times [0, T]$, and denote by (Markovian) policy a function $\pi(X_t, t) \in \mathbb{P}(\mathcal{A})$ mapping a state $(x, t) \in \mathcal{X}$ to a density over the action space $\mathcal{A}$, and denote with $p_t^{\pi}$ the marginal density at time $t$ induced by policy $\pi$. In particular, we will consider deterministic policies so that $a_t = \pi(X_t, t)$.

**Pre-trained diffusion model as an RL policy.** A pre-trained diffusion model with score function $s^{pre}$ can be interpreted as an action process $a_t^{pre} := s^{pre}(X_t^{\leftarrow}, T-t)$, where $a_t^{pre}$ is sampled from a continuous-time RL policy $a_t^{pre} \sim \pi^{pre}$. As a consequence, we can express the backward SDE induced by the pre-trained score $s^{pre}$ as follows:

$$dX_t^{\leftarrow} = b(X_t^{\leftarrow}, t, a_t^{pre})dt + \eta\sigma(t)dB_t \quad (4)$$

where we define $b(x, t, a) := -f(x, T-t) + \frac{1+\eta^2}{2}g^2(T-t) \cdot a$ and $\sigma(t) = g(T-t)$ (Zhao et al., 2024). In the following, we denote the pre-trained diffusion model by its (implicit) policy $\pi^{pre}$, which induces a marginal density $p_T^{pre} := p_T^{\pi^{pre}}$ approximating the data distribution $p_{data}$.

## 3. Problem Setting: Maximum Entropy Manifold Exploration

In this work, we aim to fine-tune a pre-trained diffusion model $\pi^{pre}$ to obtain a new model $\pi^*$, inducing a process:

$$d\overline{X}_t = b(\overline{X}_t, t, a_t^*)dt + \eta\sigma(t)dB_t \text{ with } a_t^* \sim \pi_t^* \quad (5)$$

that rather than imitating the data distribution $p_{data}$ aims to induce a marginal state distribution $p_T^{\pi^*}$ that maximally explores the *approximate data manifold* $\Omega_{pre}$ defined as:

$$\Omega_{pre} = \text{supp}(p_T^{pre}) \quad (6)$$

Formally, we pose the exploration problem as optimization of an entropy functional over the space of marginal distributions $p_T^{\pi}$ supported over the approximate data manifold, as shown in Figure 2a. Crucially, $\Omega_{pre}$, which is typically a complex set, e.g., a molecular space, is defined only implicitly via the pre-trained diffusion model $\pi^{pre}$ as expressed in Equation (6). Formally, we state the exploration problem as follows.

---

**Maximum Entropy Manifold Exploration**

$$\arg\max_{\pi} \quad \mathcal{H}(p_T^{\pi}) \quad (7)$$

$$s.t. \quad p_T^{\pi} \in \mathbb{P}(\Omega_{pre})$$

---

In this formulation, $\mathcal{H} \in \mathbb{F}(\Omega_{pre})$ denotes the differential entropy functional quantifying exploration, expressed as:

$$\mathcal{H}(\mu) = -\int d\mu \log \frac{d\mu}{dx}, \quad \mu \in \mathbb{P}(\Omega_{pre}) \quad (8)$$

For this objective to be well defined, i.e., the maximum is achieved by some measure $\mu \in \mathbb{P}(\Omega_{pre})$, a sufficient condition is stated in the following and proved in Appendix A.

**Proposition 1** ($\Omega_{pre}$ is compact). *Suppose that $s^{pre}$ is Lipschitz and the noise distribution $p_0$ is chosen as the truncated Gaussian. Then $\Omega_{pre}$ spanned by an ODE sampler is compact.*

Notice that the assumptions in Proposition 1 are standard for analysis of diffusion processes, (e.g., Lee et al., 2022; Pidstrigach, 2022), and not limiting in practice. Proposition 1 implies that $\Omega_{pre}$ is a bounded subset of $\mathbb{R}^d$, which according to the manifold hypothesis approximates a lower-dimensional data manifold (Li et al., 2024; Chen et al., 2023; Stanczuk et al., 2024; Kamkari et al., 2024), as illustrated in Figure 1.

Crucially, both the constraint set $\mathbb{P}(\Omega_{pre})$ and the marginal density $p_T^\pi$ in Problem 7 are *never represented explicitly*, but only implicitly as functions of the pre-trained policy $\pi^{pre}$ and of the new policy $\pi$ respectively.

In the rest of this work, we show that Problem (7) can be solved by fine-tuning the initial pre-trained model with respect to rewards obtained by sequentially linearizing the entropy functional. Towards this goal, in the next section, we introduce a scalable algorithmic principle that guides exploration by leveraging the representational capacity of the pre-trained diffusion model.

# 4. A Principle for Scalable Exploration

As a first step towards tackling the maximum entropy manifold exploration problem in Equation (7), we introduce a principle for exploration corresponding to a specific (intrinsic) reward function for fine-tuning.

To this end, we define the first variation of a functional over a space of probability measures (Hsieh et al., 2019). A functional $\mathcal{F} \in \mathbb{F}(\mathcal{X})$, where $\mathcal{F} : \mathbb{P}(\mathcal{X}) \to \mathbb{R}$, has first variation at $\mu \in \mathbb{P}(\mathcal{X})$ if there exists a function $\delta\mathcal{F}(\mu) \in \mathbb{F}(\mathcal{X})$ such that for all $\mu' \in \mathbb{P}(\mathcal{X})$ it holds that:

$$\mathcal{F}(\mu + \epsilon\mu') = \mathcal{F}(\mu) + \epsilon\langle\mu', \delta\mathcal{F}(\mu)\rangle + o(\epsilon).$$

where the inner product is interpreted as an expectation.

We can now present the following exploration principle as KL-regularized fine-tuning of the pre-trained model to maximize the entropy first variation evaluated at $p_T^{pre}$.

---

**Regularized Entropy First Variation Maximization**

$$\arg\max_{\pi} \quad \langle\delta\mathcal{H}\left(p_T^{pre}\right), p_T^\pi\rangle - \alpha D_{KL}(p_T^\pi, p_T^{pre}) \quad (9)$$

---

## 4.1. Generative exploration via density estimation

Crucially, this algorithmic principle does not rely on explicit uncertainty quantification and uses the generative model's ability to represent the density $p_T^{pre}$ to direct exploration. By introducing a function $f : \mathcal{X} \to \mathbb{R}$ defined for all $x \in \mathcal{X}$ as:

$$f(x) := \delta\mathcal{H}\left(p_T^{pre}\right)(x) = -\log\left(p_T^{pre}\right)(x) \quad (10)$$

the exploration principle in Equation (9) computes a policy $\pi^*$ inducing $p_T^{\pi^*}$ with high density in regions where

$p_T^{pre}$ has low density due to limited pre-training samples. Moreover, the KL regularization in Equation (9) implicitly enforces $p_T^{\pi^*}$ to lie on the approximate data manifold $\Omega_{pre}$. Formally, we have that:

$$\Omega_{\pi^*} := \text{supp}(p_T^{\pi^*}) \subseteq \text{supp}(p_T^{pre}) = \Omega_{pre} \quad \forall\alpha > 0 \quad (11)$$

The entropy first variation in Equation (10) can be interpreted as a measure of *surprise*, while the entropy functional in Equation (7) as expected surprise (Achiam & Sastry, 2017).

## 4.2. Easy to optimize, but hard to estimate density

Existing fine-tuning methods for diffusion models can only optimize linear functionals of $p_T^\pi$, namely $\mathcal{L}(\mu) = \langle f, \mu\rangle \in \mathbb{F}(\mathcal{X})$, since they can be represented as classic (reward) functions $f : \mathcal{X} \to \mathbb{R}$, defined over the design space $\mathcal{X}$, e.g., space of molecules. Although the entropy functional $\mathcal{H}$ in Equation (7) is non-linear with respect to $p_T^\pi$, its first variation is a linear functional. As a consequence, by rewriting it as shown in Equation (10), it can be optimized using existing fine-tuning methods for classic reward functions via stochastic optimal control schemes (e.g., Uehara et al., 2024b; Domingo-Enrich et al., 2024; Zhao et al., 2024), where the fine-tuning objective is:

$$\pi^* \in \arg\max_{\pi} \mathbb{E}_{x\sim\pi}\left[-\log\left(p_T^{pre}\right)(x)\right] - \alpha D_{KL}(p_T^\pi, p_T^{pre})$$

We have shown that exploration can be self-guided by a generative model using its representational power of the density it induces. But unfortunately, estimating this quantity (i.e., $p_T^{pre}$) is well-known to be a challenging task in real-world high-dimensional settings (Song et al., 2020; Kingma et al., 2021; Skreta et al., 2024).

## 4.3. Generative exploration without density estimation

In the following, we show that in the case of diffusion models, the entropy's first variation at the marginal density $p_T^\pi$ induced by $\pi$, as in Equation (10) with $\pi = \pi^{pre}$, can be optimized fully bypassing density estimation. This can be achieved by leveraging the following fundamental connection between the gradient of the entropy first variation $\nabla_x\delta\mathcal{H}\left(p_T^\pi\right)$ and the score $s^\pi(\cdot, T)$.

---

**Gradient of entropy first variation = Negative score**

$$\nabla_x\delta\mathcal{H}\left(p_T^\pi\right) = -\nabla_x\log p_T^\pi \simeq -s^\pi(\cdot, T) \quad (12)$$

---

Using Equation (12), it is possible to solve the maximization problem in Equation (9) by leveraging a first-order fine-tuning method such as Adjoint Matching (Domingo-Enrich et al., 2024) with $\nabla_x f(x) := -s^{pre}(x, T)$ as reward

gradient, where $s^{pre}$ is the known (neural) score model approximating the true score function. This realization overcomes the limitation of density estimation and renders the method scalable for high-dimensional real-world problems. For the sake of completeness, we report a detailed pseudocode of its implementation in Appendix D.

### 4.4. Beyond maximum entropy exploration

As shown in Section 8, beyond the goal of maximum (entropy) exploration, the principle in Equation (9) can be used to achieve the desired trade-off between exploration and validity by controlling the regularization coefficient $\alpha$. Higher $\alpha$ values lead to a fine-tuned model $\pi$ that conservatively aligns with the validity encoded in the pre-trained model $\pi^{pre}$. In contrast, low $\alpha$ values enable exploration of low density regions within the approximate data manifold $\Omega_{pre}$, as illustrated in Figure 2b. The latter modality is particularly relevant when a validity checker is available, e.g., synthetic accessibility (SA) scores for molecules (Ertl & Schuffenhauer, 2009), or formal verifiers for logic circuits (Coudert & Madre, 1990), allowing the discovery of new valid designs that expand the current manifold or dataset, effectively performing a guided data augmentation process (Zheng et al., 2023).

In particular, one might wonder if there exists a value of $\alpha$ such that the obtained fine-tuned model can provably solve the maximum entropy exploration problem in Equation (7). In the following section, we present a theoretical framework that answers this question positively under the idealized assumptions of exact score estimation and optimization oracle.

## 5. Provably Optimal Exploration in One Step

In this section, we show that under the assumptions of exact optimization and estimation of the entropy first variation $\delta\mathcal{H}\left(p_T^{pre}\right)$, a single fine-tuning step using Equation (9) yields an optimally explorative policy $\pi$ for entropy maximization over $\Omega_{pre}$. Complete proofs are reported in Appendices B and C.

We start by recalling the notion of Bregman divergence induced by a functional $\mathcal{Q} \in \mathbb{F}(\mathcal{X})$ between two densities $\mu, \nu \in \mathbb{P}(\mathcal{X})$, namely:

$$D_{\mathcal{Q}}(\mu, \nu) := \mathcal{Q}(\mu) - \mathcal{Q}(\nu) - \langle \delta\mathcal{Q}(\nu), \mu - \nu \rangle$$

Next, we introduce two structural properties for our analysis.[2]

**Definition 1** (Relative smoothness and relative strong convexity (Lu et al., 2018)). *Let $\mathcal{F} : \mathbb{P}(\mathcal{X}) \to \mathbb{R}$ a convex*

functional. We say that $\mathcal{F}$ is L-smooth relative to $\mathcal{Q} \in \mathbb{F}(\mathcal{X})$ over $\mathbb{P}(\mathcal{X})$ if $\exists\, L$ scalar s.t. for all $\mu, \nu \in \mathbb{P}(\mathcal{X})$:

$$\mathcal{F}(\nu) \leq \mathcal{F}(\mu) + \langle \delta\mathcal{F}(\mu), \nu - \mu \rangle + LD_{\mathcal{Q}}(\nu, \mu) \quad (13)$$

and we say that $\mathcal{F}$ is l-strongly convex relative to $\mathcal{Q} \in \mathbb{F}(\mathcal{X})$ over $\mathbb{P}(\mathcal{X})$ if $\exists\, l \geq 0$ scalar s.t. for all $\mu, \nu \in \mathbb{P}(\mathcal{X})$:

$$\mathcal{F}(\nu) \geq \mathcal{F}(\mu) + \langle \delta\mathcal{F}(\mu), \nu - \mu \rangle + lD_{\mathcal{Q}}(\nu, \mu) \quad (14)$$

In the following, we view the principle in Equation (9) as a step of mirror descent (Nemirovskij & Yudin, 1983) and the KL divergence term as the Bregman divergence induced by an entropic mirror map $\mathcal{Q} = -\mathcal{H}$, i.e., $D_{KL}(\mu, \nu) = D_{-\mathcal{H}}(\mu, \nu)$. We can now state the following lemma regarding $\mathcal{F} = -\mathcal{H}$.

**Lemma 5.1** (Relative smoothness and strong convexity for $\mathcal{F} = \mathcal{Q} = \mathcal{H}$). *For $\mathcal{F} = \mathcal{Q} = -\mathcal{H}$ as in Equation (9), we have that $\mathcal{F}$ is 1-smooth (i.e., $L = 1$) and 1-strongly convex (i.e., $l = 1$) relative to $\mathcal{Q}$.*

We can finally state the following set of idealized assumptions as well as the one-step convergence guarantee.

**Assumption 5.1** (Exact estimation and optimization). *We consider the following assumptions:*

1. *Exact score estimation: $s^{pre}(\cdot, T) = \nabla_x \log p_T^{pre}$*

2. *The optimization problem in Equation (9) is solved exactly.*

**Theorem 5.2** (One-step convergence). *Given Assumptions 5.1, fine-tuning a pre-trained model $\pi^{pre}$ according to Equation (9) with $\alpha = L = 1$, leads to a policy $\pi$ inducing a marginal distribution $p_T^\pi \in \mathbb{P}(\Omega_{pre})$ such that:*

$$\mathcal{H}(p_T^*) - \mathcal{H}(p_T^\pi) \leq \frac{L - l}{K} D_{KL}(p_T^*, p_T^{pre}) = 0 \quad (15)$$

*where $p_T^* := p_T^{\pi^*}$ is the marginal distribution induced by the optimal exploratory policy $\pi^* \in \arg\max_{\pi \in \Lambda} \mathcal{H}(p_T^\pi)$ with $\Lambda = \{\pi : p_T^\pi \in \mathbb{P}(\Omega_{pre})\}$ being the set of policies compatible with the approximate data manifold $\Omega_{pre}$.*

Theorem 5.2 hints at promising performances of using Equation (9) as a fine-tuning objective for maximum entropy exploration, as under Assumptions 5.1, it provably leads to an optimally explorative policy in one step. However, Assumptions 5.1 clearly do not hold in most realistic settings as:

1. The estimation quality of the score $s^{pre}(\cdot, T)$ learned from data is approximate.

---

[2]In line with standard notations in the optimization literature, we present the framework as $\min_{p_T^\pi \in \mathbb{P}(\Omega_{pre})} -\mathcal{H}(p_T^\pi)$, which is clearly equivalent to (7).

2. The high-dimensional stochastic optimal control methods used to solve Equation (9) are approximate in practice.

Therefore, optimizing the entropy first variation as in Equation (9) is actually unlikely to lead to an optimally explorative diffusion model, as shown experimentally in Section 8. To address this issue, in the next section we propose an exploration algorithm by building on the exploration principle in Equation (9).

## 6. Algorithm: Score-based Maximum Entropy Manifold Exploration

In the following, we present **S**core-based **M**aximum **E**ntropy **M**anifold **E**xploration (S-MEME), see Algorithm 1, which reduces manifold exploration to sequential fine-tuning of the pre-trained diffusion model $\pi^{pre}$ by following a mirror descent (MD) scheme (Nemirovskij & Yudin, 1983). Crucially, each iteration $k$ of S-MEME corresponds to a fine-tuning step according to Equation (9), where the pre-trained model $\pi^{pre} =: \pi_0$ is then replaced by the model at the previous iteration, namely $\pi_{k-1}$. Concretely, this makes it possible to reduce the optimization of the entropy functional, which is non-linear w.r.t. $p_T^\pi$, to a sequence of optimization problems of linear functionals.

---

**Algorithm 1** **S**core-based **M**aximum **E**ntropy **M**anifold **E**xploration (S-MEME)

---

**input** $K$ : number of iterations, $\pi^{pre}$ : pre-trained diffusion, $\{\alpha_k\}_{k=1}^K$ regularization coefficients
1: **Init:** $\pi_0 := \pi^{pre}$
2: **for** $k = 1, 2, \ldots, K$ **do**
3:     Set: $\nabla_x f_k = -s^{k-1}$ with $s^{k-1} = s^{\pi_{k-1}}$
4:     Compute $\pi_k$ via first-order linear fine-tuning:

$$\pi_k \leftarrow \text{LINEARFINETUNINGSOLVER}(\nabla_x f_k, \alpha_k, \pi_{k-1})$$

5: **end for**
**output** policy $\pi := \pi_K$

---

Algorithm 1 requires as inputs a pre-trained diffusion model $\pi^{pre}$, the number of iterations $K$, and a schedule of regularization coefficients $\{\alpha_k\}_{k=1}^K$. At each iteration, S-MEME sets the gradient of the entropy first variation evaluated at the previous policy $\pi_{k-1}$, namely $\nabla_x \delta\mathcal{H}\left(p_T^{k-1}\right)$, to be the score $s^{k-1} := s^{\pi_{k-1}}$ associated to the diffusion model $\pi_{k-1}$ obtained at the previous iteration (line 3). Then, it computes policy $\pi_k$ by solving the following fine-tuning problem

$$\arg\max_\pi \quad \mathbb{E}_{x\sim\pi}\left[-\log\left(p_T^{k-1}\right)(x)\right] - \alpha_k KL(p_T^\pi, p_T^{k-1})$$

via a first-order solver such as Adjoint Matching (Domingo-Enrich et al., 2024), using $\nabla f_k := -s^{k-1}(\cdot, T)$ as in

Eq. (12) (line 4). Ultimately, it returns a final policy $\pi := \pi_K$. We report a possible implementation of LINEARFINETUNINGSOLVER in Appendix D.

Crucially, S-MEME controls the distributional behavior of the final diffusion model $\pi$, which is essential to optimize the entropy as it is a non-linear functional over $\mathbb{P}(\Omega_{pre})$.

However, it is still unclear whether the algorithm provably converges to the optimally explorative diffusion model $\pi^*$. In the next section, we answer affirmatively this question by developing a theoretical analysis under general assumptions based on recent results for mirror flows (Hsieh et al., 2019).

## 7. Manifold Exploration Guarantees

The purpose of this section is to establish a realistic framework under which Algorithm 1 is guaranteed to solve the maximum entropy manifold exploration Problem (7).

### 7.1. Key Assumptions

We now present all the assumptions and provide an explanation of why they are realistic. Conceptually, these assumptions align with the *stochastic approximation* framework of Benaïm (2006); Mertikopoulos et al. (2024); Hsieh et al. (2021). Specifically, recall that $p_T^k := p_T^{\pi_k}$ represents the (stochastic) density produced by the LINEARFINETUNING-SOLVER oracle at the $k$-th step of S-MEME, and consider the following *mirror descent* iterates:

$$p_\sharp^k := \arg\max_{p\in\mathbb{P}(\Omega_{pre})} \langle d\mathcal{H}\left(p_T^{\pi_{k-1}}\right), p\rangle - \frac{1}{\gamma_k}D_{KL}(p, p_T^{\pi_{k-1}}) \tag{MD$_k$}$$

where $1/\gamma_k = \alpha_k$ in Algorithm 1.

As explained in Section 5, the maximum entropy manifold exploration problem (7) can be solved in a single step using (MD$_k$). However, in realistic settings where only noisy *and* biased approximations of (MD$_k$) are available, it becomes essential to quantify the deviations due to these approximations from the idealized iterates in (MD$_k$). Additionally, the step sizes $\gamma_k$ must be carefully designed to account for such deviations. This section aims to achieve precisely this goal. To this end, we first require:

**Assumption 7.1** (Support Compatibility). *We assume that* $supp(p_T^{\pi_k}) \subset \tilde\Omega$ *for all* $k$, *and* $supp(p_j^{\pi_k}) = \tilde\Omega$ *for some* $j$.

Next, we require a purely technical assumption that is typically satisfied in practice:

**Assumption 7.2.** *The sequence* $\{\delta\mathcal{H}(p_T^{\pi_k})\}_k$ *is precompact in the topology induced by the* $L_\infty$ *norm.*

Now, denote by $\mathcal{G}_k$ the filtration up to step $k$, and consider the decomposition of the oracle into its *noise* and *bias* parts:

$$b_k := \mathbb{E}\left[\delta\mathcal{H}(p_T^{\pi_k}) - \delta\mathcal{H}(p_\sharp^k) \,|\, \mathcal{G}_k\right] \tag{16}$$

$$U_k := \delta\mathcal{H}(p_T^{\pi_k}) - \delta\mathcal{H}(p_\sharp^k) - b_k \tag{17}$$

Observe that, conditioned on $\mathcal{G}_k$, $U_k$ has zero mean, while $b_k$ captures the *systematic* error. We then impose the following:

**Assumption 7.3** (Noise and Bias). *The following events happen almost surely:*

$$\|b_k\|_\infty \to 0 \tag{18}$$

$$\sum_k \mathbb{E}\left[\gamma_k^2\left(\|b_k\|_\infty^2 + \|U_k\|_\infty^2\right)\right] < \infty \tag{19}$$

$$\sum_k \gamma_k \|b_k\|_\infty < \infty \tag{20}$$

Two important remarks are worth noting. First, (18) represents a *necessary* condition for convergence: if this condition is violated, it becomes straightforward to construct examples where no practical algorithm can successfully solve the maximum entropy problem. Second, (19) and (20) address the trade-off between the *accuracy* of the approximate oracle LINEARFINETUNINGSOLVER and the aggressiveness of the step sizes, $\gamma_k$. Intuitively, a smaller noise and bias allows for the use of larger step sizes. In this context, (19) and (20) establish a concrete criterion ensuring that the task of finding the optimally explorative policy succeeds with probability 1.

We are now finally ready to state the following result.

**Theorem 7.1** (Convergence under general assumptions). *Consider the standard Robbins-Monro step-size rule: $\sum_k \gamma_k = \infty, \sum_k \gamma_k^2 < \infty$. Then under Assumptions 7.1 to 7.3, the sequence of marginal densities $p_T^k$ induced by the iterates $\pi_k$ of Algorithm 1 converges weakly to $p_T^*$ almost surely. Formally, we have that:*

$$p_T^k \rightharpoonup p_T^* \quad a.s. \tag{21}$$

*where $p_T^* \in \arg\max_{p_T \in \mathbb{P}(\Omega_{pre})} \mathcal{H}(p_T)$ is the maximum entropy marginal density compatible with $\Omega_{pre}$.*

**Remark.** It is possible to derive an explicit convergence rate for Theorem 7.1 corresponding to $\tilde{\mathcal{O}}((\log\log k)^{-1})$, which is, in general, the best achievable without additional assumptions (Karimi et al., 2024). We omit the technical details, as they are rather involved and offer limited practical relevance.

## 8. Experimental Evaluation

In this section, we analyze the ability of S-MEME to induce explorative policies on two tasks: (1) An illustrative example to showcase visually interpretable exploration and ability to sample from low-density regions (see Figure 3), and (2) A text-to-image task aiming to explore the approximate manifold of *creative architecture* designs (see Figure 4). Additional details on experiments are provided in Appendix E.

**(1) Illustrative setting.** In this experiment, we consider

the common setting where the density $p_T^{pre}$ induced by a pre-trained model $\pi^{pre}$ presents a high-density region (yellow area in Figure 3a) and a low-density region (green area in in Figure 3a). As illustrated in Figure 3a, the pre-trained model $\pi^{pre}$ induces an unbalanced density, where $N = 80000$ samples are obtained mostly from the high-density area. For quantitative evaluation, we compute a Monte Carlo estimate of $\mathcal{H}(p_T^\pi)$. Crucially, Figure 3 shows that S-MEME can induce a highly explorative density in terms of entropy (see Figure 3d), compared with the pre-trained model, after only $K = 4$ iterations. One can notice that the density induced by the fine-tuned model (see Figure 3c) is significantly more uniform and higher in low-probability regions for the pre-trained model (see right region of Figure 3c), while preserving the support of the data distribution.

| | $\mathbf{p_T^{pre}}$ | **S-MEME 1** | **S-MEME 2** | **S-MEME 3** |
|---|---|---|---|---|
| **FID**$(\mathbf{p}, \mathbf{p_T^{pre}})$ | 0.0 | 10.25 | 9.83 | 19.15 |
| **CLIP** | 22.27 | 20.79 | 20.88 | 19.86 |
| $\widehat{\mathcal{H}}(\mathbf{p}, \mathbf{p_T^{pre}})$ | -1916.47 | 564.72 | 482.81 | 843.88 |

Table 1: FID, CLIP and cross-entropy evaluation of $p_T^{pre}$ and $p_T^{\pi_k}$. For $k = 1, 2, 3$, S-MEME achieves larger distance to $p_T^{pre}$ while preserving high CLIP score.

**(2) Text-to-image manifold exploration.** We consider the problem of exploring the data manifold of *creative architecture* designs given a pre-trained text-to-image diffusion model. For this we utilize the stable diffusion (SD) 1.5 (Rombach et al., 2021) checkpoint pre-trained on the LAION-5B dataset (Schuhmann et al., 2022). Since SD-1.5 uses classifier-free guidance (Ho & Salimans, 2022), we fine-tune the velocity resulting from applying the classifier-free guidance with a guidance scale of $w = 8$ which is standard for SD-1.5. Similarly, we also use the same guidance scheme for the fine-tuned model. We fine-tuned the checkpoint with $K = 3$ iterations of S-MEME on a single Nvidia H100 GPU for the prompt "A creative architecture.". In Figure 4, we show images generated from $\pi^{pre}$, $\pi_1$ and $\pi_3$, resulting from the same initial noise samples. One can notice an increase in the complexity and originality of the respective images, likely hinting at higher probability of the fine-tuned model to sample from a lower-density region for $\pi^{pre}$. Moreover, less conservative architectures are sampled with more steps of S-MEME while preserving semantic faithfulness. We measure this in Table 1 by computing the Fréchet inception distance (FID) (Heusel et al., 2017) and Gaussian cross-entropy in feature space of Inception-v3 between $p_T^{\pi_k}$ and $p_T^{pre}$ for $k = 1, 2, 3$ as well as the CLIP score (Hessel et al., 2021) for the distribution induced by the specific prompt. The main reason for these proxy metrics is the intractability of computing $\log p_T^{pre}(x)$ in high-dimensional spaces, such as that of images generated by a large diffusion model. One can notice from Table 1 an increase in FID and cross-entropy between the distributions as $k$ increases,

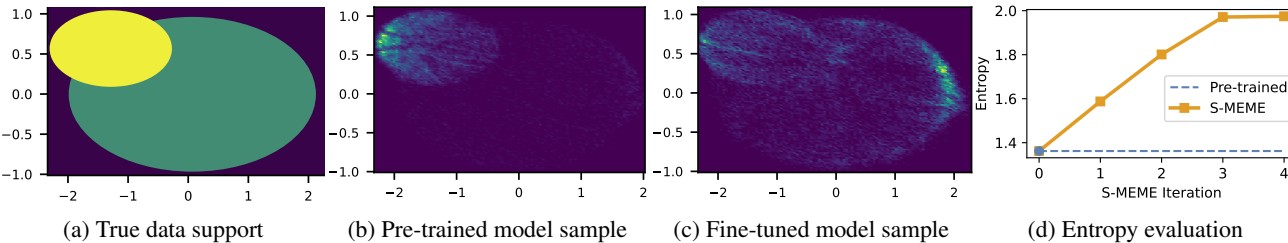

(a) True data support  (b) Pre-trained model sample  (c) Fine-tuned model sample  (d) Entropy evaluation

Figure 3: Illustrative example with unbalanced pre-trained model $\pi^{pre}$. (3a) Support of true data distribution composed of a small high-density region (yellow) and a wide low-density area (green). (3b) Sample from pre-trained model $\pi^{pre}$. (3c) Sample from $\pi_4$ obtained after 4 steps of S-MEME. (3d) Entropy estimation of densities $\{p_k\}_{k=1}^{K}$ obtained via $K = 4$ steps. Notice that S-MEME returns a fine-tuned model with significantly higher entropy than $\pi^{pre}$ (see 3c and 3d), and higher density in low-density regions for the pre-trained model (compare (3a) and (3c)), while preserving the data support shown in 3a.

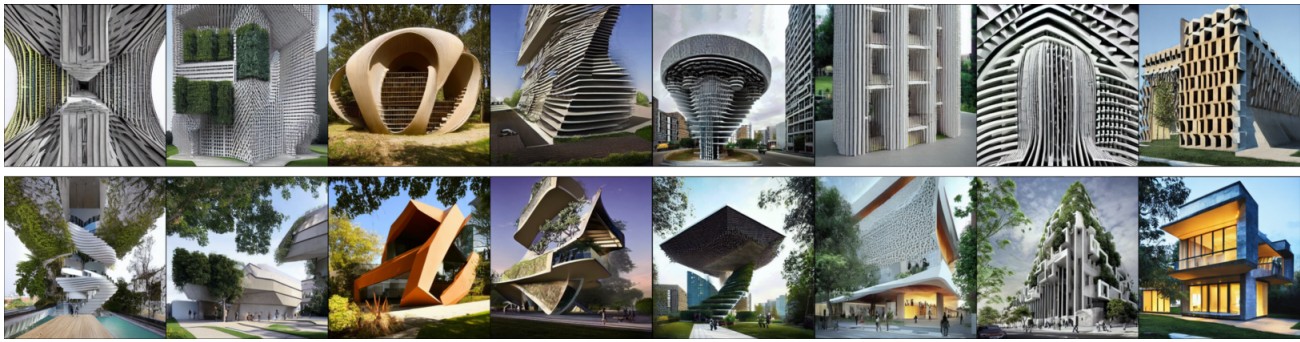

Figure 4: Generated images from $\pi^{pre}$ (top) and $\pi_3$ (bottom) for a fixed set of initial noisy samples using the prompt "A creative architecture.". We observe an increase in complexity and originality of the S-MEME generated images while preserving semantic faithfulness, likely hinting at higher probability of sampling from a lower-density region of $\pi^{pre}$.

while the fine-tuned model preserves the CLIP score of the pre-trained model. We provide further results for text-to-image in Appendix G, and in Appendix F, we evaluate the Vendi score (Friedman & Dieng, 2022) as a diversity metric.

## 9. Related Work

In the following, we present relevant work in related areas.

**Maximum State Entropy Exploration.** Maximum state entropy exploration, introduced by Hazan et al. (2019), addresses the pure-exploration RL problem of maximizing the entropy of the state distribution induced by a policy over a dynamical system's state space (e.g., Lee et al., 2019; Mutti et al., 2021; Guo et al., 2021). The presented manifold exploration problem is closely related, with $p_T^{\pi}$ representing the state distribution induced by policy $\pi$ over a subset of the state space. Nonetheless, in Problem (7) the admissible state distributions $\mathbb{P}(\Omega_{pre})$ are represented only implicitly via a pre-trained generative model $\pi^{pre}$, able to capture complex design spaces, e.g., valid molecules. Moreover, exploration is guided by the diffusion model's score function via Eq. 12, overcoming the need of explicit entropy or density estimation, a fundamental challenge

in this area (Liu & Abbeel, 2021; Seo et al., 2021; Mutti et al., 2021). Recent studies have tackled maximum entropy exploration with finite sample budgets (e.g., Mutti et al., 2022b;a; 2023; Prajapat et al., 2023; De Santi et al., 2024b). We believe several ideas presented in this work can extend to such settings. Ultimately, to the best of our knowledge, this is the first work providing a rigorous theoretical analysis of maximum state entropy exploration over continuous state spaces, albeit for a specific sub-case, as well as leveraging this formulation for fine-tuning of diffusion models.

**Continuous-time RL.** Continuous-time RL extends stochastic optimal control (Fleming & Rishel, 2012) to handle unknown rewards or dynamics (e.g., Doya, 2000; Wang et al., 2020). Problem (7) represents the pure exploration case of continuous-time RL, were the goal is to compute a (purely) exploratory policy $\pi$ over a subset of the state space $\Omega_{pre} \subseteq \mathcal{X}$ implicitly defined by a pre-trained generative model $\pi^{pre}$. Moreover, Problem (7) can be further motivated as a continuous-time RL reward learning setting (e.g., Lindner et al., 2021; Mutny et al., 2023; De Santi et al., 2024a), where an agent aims to learn an unknown homoscedastic reward function such as toxicity over a molecular space (Yang et al., 2022). To our

knowledge, this is the first work that tackles maximum entropy exploration in a continuous-time RL setting.

**Diffusion models fine-tuning via optimal control.** Recent works have framed diffusion models fine-tuning with respect to a reward function $f : \mathcal{X} \to \mathbb{R}$ as an entropy-regularized stochastic optimal control problem (e.g., Uehara et al., 2024a; Tang, 2024; Uehara et al., 2024b; Domingo-Enrich et al., 2024). In this work, we introduce a scalable fine-tuning scheme, based on first-order solvers for classic rewards (e.g., Domingo-Enrich et al., 2024), that optimizes a broader class of functionals requiring information about the full density $p_T^\pi$, such as entropy and alternative exploration measures (De Santi et al., 2024b; Hazan et al., 2019). This paves the way to using diffusion models for optimization of distributional objectives, rather than simple scalar rewards. Beyond classic optimization, our framework is particularly relevant for Bayesian optimization, or bandit, problems (e.g., Uehara et al., 2024b), where the reward function to be optimized over the manifold is unknown and therefore exploration is essential.

**Sample diversity in diffusion models generation.** The lack of sample diversity in diffusion model generation is a key challenge tackled by various works (e.g., Corso et al., 2023; Um et al., 2023; Kirchhof et al., 2024; Sadat et al., 2024; Um & Ye, 2025). These methods complement ours by enabling diverse sampling from the fine-tuned explorative model obtained via S-MEME. While prior works focus on generating diverse samples from a fixed diffusion model, ours provides a framework for manifold exploration as policy optimization via reinforcement learning. This enables scalable and provable maximization of typical exploration measures in RL, such as state entropy (Hazan et al., 2019). Among related works, Miao et al. (2024) shares the closest intent, but lacks a formal setting with exploration guarantees, and the diffusion model's exploration process relies on computing an external metric for exploration, rather than being self-guided via its own score function as S-MEME achieves via Eq. (12).

**Optimization over probability measures via mirror flows.** Recently, there has been a growing interest in analyzing optimization problems over spaces of probability measures. Existing works have explored applications including GANs (Hsieh et al., 2019), optimal transport (Aubin-Frankowski et al., 2022; Léger, 2021; Karimi et al., 2024), and kernelized methods (Dvurechensky & Zhu, 2024). However, to the best of our knowledge, the case of the entropy-based objective in Problem 7, along with its associated relative smoothness analysis framework, has not been previously addressed. Moreover, prior approaches do not leverage a key aspect of our framework: the admissible space of probability measures, $\mathbb{P}(\Omega_{pre})$, is represented only implicitly via a pre-trained generative model, which can approximate complex data manifolds learned from data, such as molecular spaces.

This idea of optimization with implicit constraints captured by generative models is novel and absent in earlier work.

## 10. Conclusion

This work tackles the fundamental challenge of leveraging the representational power of generative models for exploration. We first introduce a formal framework for exploration as entropy maximization over the approximate data manifold implicitly captured by a pre-trained diffusion model. Then, we present an algorithmic principle that guides exploration via density estimation, a challenging task in real-world settings. By exploiting a fundamental connection between entropy and a diffusion model's score function, we overcome this problem and ensure scalability of the proposed principle for exploration. Building on this, we introduce S-MEME, a sequential fine-tuning algorithm that provably solves the exploration problem, with convergence guarantees grounded in recent advances in mirror flows. Finally, we validate the proposed method on both a conceptual benchmark and a high-dimensional text-to-image task, demonstrating its practical relevance.

## Acknowledgements

This publication was made possible by the ETH AI Center doctoral fellowship to Riccardo De Santi, and postdoctoral fellowship to Marin Vlastelica. The project has received funding from the Swiss National Science Foundation under NCCR Catalysis grant number 180544 and NCCR Automation grant agreement 51NF40 180545.

## Impact Statement

In this work we provide a method and theoretical analysis of maximum-entropy exploration for diffusion generative models as well as an implemented algorithm. To the best of our knowledge, this will enable further more applied research in this areas with exciting applications.

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

# A. Proof for Proposition 1

Recall the probability flow ODE in (Song et al., 2020, eq. (13)), which is what we use to generate $p_T^{pre}$ (this is also a common practice in the literature). We know that the generative ODE corresponding to the backward SDE Equation (4) is written as (suppose $\eta = 1$ for simplicity)

$$\mathrm{d}X_t^{\leftarrow} = \underbrace{-f(X_t^{\leftarrow}, T-t) + \frac{1}{2}g^2(T-t)s^{pre}(X_t^{\leftarrow}, T-t)}_{:=v(X_t^{\leftarrow}, t)} \mathrm{d}t.$$

Here $f$ is defined in the forward process Equation (1). Clearly the velocity field $v(x, t)$ is Lipschitz w.r.t. $x$ due to the assumption that $s^{pre}$ is Lipschitz and the fact that $f$ is linear w.r.t. $x$. Consequently, *the flow map induced by the above ODE is Lipschitz*. As a result, since $X_0$ is sampled from a truncated Gaussian distribution which has a compact support, $\Omega^{pre} = \mathrm{supp}(p_T^{pre})$ is also compact for any finite $T$.

# B. Proof for Section 5

**Theorem 5.2** (One-step convergence). *Given Assumptions 5.1, fine-tuning a pre-trained model $\pi^{pre}$ according to Equation (9) with $\alpha = L = 1$, leads to a policy $\pi$ inducing a marginal distribution $p_T^\pi \in \mathbb{P}(\Omega_{pre})$ such that:*

$$\mathcal{H}(p_T^*) - \mathcal{H}(p_T^\pi) \leq \frac{L-l}{K} D_{KL}(p_T^*, p_T^{pre}) = 0 \tag{15}$$

*where $p_T^* := p_T^{\pi^*}$ is the marginal distribution induced by the optimal exploratory policy $\pi^* \in \arg\max_{\pi \in \Lambda} \mathcal{H}(p_T^\pi)$ with $\Lambda = \{\pi : p_T^\pi \in \mathbb{P}(\Omega_{pre})\}$ being the set of policies compatible with the approximate data manifold $\Omega_{pre}$.*

*Proof.* Towards proving this result, we interpret Eq. (9) as the first iteration of Algorithm 1. Hence, to prove the statement, it is sufficient to show that Algorithm 1 after one iteration computes $\pi_1$ inducing density $p_T^{\pi_1}$ such that $\mathcal{H}(p_T^*) = \mathcal{H}(p_T^\pi)$. We prove this result by leveraging the properties of relative smoothness and relative strong convexity introduced in Sec. 5.

The analysis is bases on a classic analysis for mirror descent via relative properties (Lu et al., 2018) First, we show the following, where for the sake of using a simple notation, we denote $p_T^{\pi_k}$ by $\mu_k$, and consider an arbitrary density $\mu \in \mathbb{P}(\Omega_{pre})$.

$$\mathcal{H}(\mu_k) \leq \mathcal{H}(\mu_{k-1}) + \langle \delta\mathcal{H}(\mu_{k-1}), \mu_k - \mu_{k-1} \rangle + LD_\mathcal{Q}(\mu_k, \mu_{k-1}) \tag{22}$$
$$\leq \mathcal{H}(\mu_{k-1}) + \langle \delta\mathcal{H}(\mu_{k-1}), \mu - \mu_{k-1} \rangle + LD_\mathcal{Q}(\mu, \mu_{k-1}) - LD_\mathcal{Q}(\mu, \mu_k) \tag{23}$$

where in the first inequality we have used the $L$-smoothness of $\mathcal{H}$ relative to $\mathcal{Q} = \mathcal{H}$ as in Definition 1, while in the last inequality we have used the three-point property of the Bregman divergence (Lu et al., 2018, Lemma 3.1) with $\phi(\mu) = \frac{1}{L}\langle \delta\mathcal{H}(\mu_{k-1}), \mu - \mu_{k-1} \rangle$, $z = \mu_{k-1}$, and $z^+ = \mu_k$. Then, we can derive:

$$\mathcal{H}(\mu_k) \leq \mathcal{H}(\mu) + (L - \mu)D_\mathcal{Q}(\mu, \mu_{k-1}) - LD_\mathcal{Q}(\mu, \mu_k) \tag{24}$$

by using the $l$-strong convexity of $\mathcal{H}$ relative to $\mathcal{Q} = \mathcal{H}$ as in Definition 1. By induction, using monotonicity of the iterates and non-negativity of the Bregman divergence as in (Lu et al., 2018), one obtains:

$$\sum_{k=1}^{K} \left(\frac{L}{L-l}\right)^k (\mathcal{H}(\mu_k) - \mathcal{H}(\mu)) \leq LD_\mathcal{Q}(\mu, \mu_0) - L\left(\frac{L}{L-l}\right)D_\mathcal{Q}(\mu, \mu_k) \leq LD_\mathcal{Q}(\mu, \mu_0) \tag{25}$$

Defining:

$$\frac{1}{C_k} = \sum_{k=1}^{K} \left(\frac{L}{L-l}\right)^k \tag{26}$$

and rearrenging the terms leads to:

$$\mathcal{H}(\mu_k) - \mathcal{H}(\mu) \leq C_k LD_\mathcal{Q}(\mu, \mu_0) = \frac{\mu D_Q(\mu, \mu_0)}{\left(1 + \frac{l}{L-l}\right)^l - 1} \tag{27}$$

Given Eq. 27, the convergence in the statement can be derived using Lemma 5.1, and the fact that $\left(1 + \frac{l}{L-l}\right)^k \geq 1 + \frac{k\mu}{L-\mu}$. Ultimately, $p_T^\pi \in \mathbb{P}(\Omega_{pre}) \forall \alpha > 0$ is trivially due to the fact that $\Omega_{pre}$ is the support of the right element of the Kullback–Leibler divergence in Eq. 9. $\square$

# C. Proof for Section 7

## C.1. Proof of Theorem 7.1

We restate the theorem for reader's convenience:

**Theorem 7.1** (Convergence under general assumptions)**.** *Consider the standard Robbins-Monro step-size rule:* $\sum_k \gamma_k = \infty, \sum_k \gamma_k^2 < \infty$. *Then under Assumptions 7.1 to 7.3, the sequence of marginal densities $p_T^k$ induced by the iterates $\pi_k$ of Algorithm 1 converges weakly to $p_T^*$ almost surely. Formally, we have that:*

$$p_T^k \rightharpoonup p_T^* \quad a.s. \tag{21}$$

*where $p_T^* \in \arg\max_{p_T \in \mathbb{P}(\Omega_{pre})} \mathcal{H}(p_T)$ is the maximum entropy marginal density compatible with $\Omega_{pre}$.*

*Proof.* To enhance the readability of our proof, we begin by outlining the key steps.

**Proof Outline.** The main idea is to analyze the convergence of the iterates $\{p_T^k\}_{k \in \mathbb{N}}$ generated by Algorithm 1 by relating them to a corresponding *continuous-time* dynamical system. Specifically, we define the initial dual variable as

$$h_0 = \delta\mathcal{H}(p_{pre}) = -\log p_{pre},$$

and consider the following system:

$$\begin{cases} \dot{h}_t = \delta\mathcal{H}(p_t) \\ p_t = \delta(-\mathcal{H})^\star(h_t) \end{cases} \equiv \begin{cases} \dot{h}_t = -\log p_t \\ p_t = \frac{e^{h_t}}{\int_\Omega e^{h_t}}. \end{cases} \tag{MF}$$

Here, $(-\mathcal{H})^\star(h) := \log \int_\Omega e^h$ is the Fenchel dual of the entropy function (Hsieh et al., 2019; Hiriart-Urruty & Lemaréchal, 2004).

To bridge the gap between discrete and continuous-time analysis, we construct a continuous-time interpolation of the discrete iterates $\{h^k\}_{k \in \mathbb{N}}$. Let $(h^k := \delta\mathcal{H}(p_T^k))_{k \in \mathbb{N}}$ be the sequence of the corresponding *dual variables*. We introduce the notion of an "effective time" $\tau^k$, defined as:

$$\tau^k := \sum_{n=1}^k \gamma_n,$$

which represents the cumulative time elapsed up to the $k$-th iteration of the discrete-time process $h^k$ using step-size $\gamma_k$. Using $\tau^k$, we define the *continuous-time interpolation $h(t)$ of $h^k$* as follows:

$$h(t) := h^k + \frac{t - \tau^k}{\tau^{k+1} - \tau^k}(h^{k+1} - h^k). \tag{Int}$$

Intuitively, the convergence of our algorithm follows if the following two conditions hold:

**Informal Assumption 1** (Closeness of discrete and continuous times)**.** *The interpolated process (Int) asymptotically approaches the continuous-time dynamics in (MF) as $k \to \infty$.*

**Informal Assumption 2** (Convergence of continuous-time dynamics)**.** *The trajectory of (MF) converges to the **optimal solution** of the maximum entropy problem (7).*

To formalize the above intuition, we leverage the *stochastic approximation* framework of Benaïm (2006); Mertikopoulos et al. (2024); Karimi et al. (2024), outlined as follows.

First, to precisely state Informal Assumption 1, we introduce a measure of "closeness" between continuous orbits. Let $\mathcal{Z}$ denote the space of integrable functions on $\Omega$ (viewed as the dual space of probability measures; see (Halmos, 2013)), and define the *flow* $\Theta \colon \mathbb{R}_+ \times \mathcal{Z} \to \mathcal{Z}$ associated with (MF). That is, for an initial condition $h_0 = h \in \mathcal{Z}$, the function $\Theta$ describes the orbit of (MF) at time $t \in \mathbb{R}_+$.

We then define the notion of "asymptotic closeness" as follows:

**Definition 2.** *We say that $h(t)$ is an asymptotic pseudotrajectory* (APT) *of* (MF) *if, for all $T > 0$, we have:*

$$\lim_{t \to \infty} \sup_{0 \le s \le T} \|h(t+s) - \Theta_s(h(t))\|_\infty = 0. \tag{28}$$

This comparison criterion, introduced by Benaïm & Hirsch (1996), plays a central role in our analysis. Intuitively, it states that $h(t)$ eventually tracks the flow of (MF) with arbitrary accuracy over arbitrarily long time windows. Consequently, if (Int) is an APT of (MF), we can reasonably expect its behavior— and thus that of $\{h^k\}_{k\in\mathbb{N}}$— to closely follow (MF).

The precise connection is established by Benaïm & Hirsch (1996) through the concept of *internally chain-transitive* (ICT) sets:

**Definition 3** (Benaïm & Hirsch, 1996; Benaïm, 2006)**.** *Let $\mathcal{S}$ be a nonempty compact subset of $\mathcal{Z}$. Then:*

1. *$\mathcal{S}$ is* invariant *if $\Theta_t(\mathcal{S}) = \mathcal{S}$ for all $t \in \mathbb{R}$.*

2. *$\mathcal{S}$ is* attracting *if it is invariant and there exists a compact neighborhood $\mathcal{K}$ of $\mathcal{S}$ such that $\lim_{t\to\infty} \mathrm{dist}(\Theta_t(h), \mathcal{S}) = 0$ uniformly for all $h \in \mathcal{K}$.*

3. *$\mathcal{S}$ is an* **internally chain-transitive** (ICT) *set if it is invariant and $\Theta|_{\mathcal{S}}$ admits no proper attractors within $\mathcal{S}$.*

The significance of ICT sets lies in (Benaïm, 2006, Theorem 5.7):

**Theorem C.1** (APTs converge to ICT sets)**.** *Let $h(t)$ be a precompact asymptotic pseudotrajectory generated by $\{h^k\}_{k\in\mathbb{N}}$ for the flow associated with the continuous-time system* (MF). *Then, almost surely, $h^k \to \mathcal{S}$, where $\mathcal{S}$ is an ICT set of* (MF).

By Theorem C.1, establishing Theorem 7.1 reduces to proving the following two statements:

1. The iterates $\{h^k\}_{k\in\mathbb{N}}$ of Algorithm 1 generate a precompact APT of (MF).

2. The unique ICT set of (MF) is the solution to the optimization problem (7).

These results provide the rigorous counterpart to Informal Assumptions 1 to 2. The proof below proceeds by formally establishing each of these points.

**The ICT set of** (MF) **is the solution to** (7)**.** By the definition (MF), we can easily see that:

$$\dot{p}_t = p_t \dot{h}_t - \frac{e^{h_t}}{\int_\Omega e^{h_t}} \cdot \frac{\int_\Omega \dot{h}_t \cdot e^{h_t}}{\int_\Omega e^{h_t}} \tag{29}$$

$$= p_t \left( \dot{h}_t - \mathbb{E}_{p_t} \dot{h}_t \right). \tag{30}$$

We then compute:

$$-\frac{\mathrm{d}}{\mathrm{d}t} \mathcal{H}(p_t) = -\langle \delta\mathcal{H}(p_t), \dot{p}_t \rangle \tag{31}$$

$$= \langle \log p_t, p_t \left( \dot{h}_t - \mathbb{E}_{p_t} \dot{h}_t \right) \rangle \qquad \text{by (30)} \tag{32}$$

$$= \int_\Omega p_t \log p_t \dot{h}_t - \int_\Omega p_t \log p_t \cdot \int_\Omega p_t \dot{h}_t \tag{33}$$

$$= -\int_\Omega p_t (\log p_t)^2 - \left( \int_\Omega p_t \log p_t \right)^2 \qquad \text{by (MF)} \tag{34}$$

$$= -\left( \mathbb{E}_{X_t \sim p_t} (\log p_t(X_t))^2 - (\mathbb{E}_{X_t \sim p_t} \log p_t(X_t))^2 \right) \tag{35}$$

$$\le 0 \tag{36}$$

by Jensen's inequality. Also, note that the inequality is strict if $h_t$ is not constant, i.e., if $p_t$ is not uniform on $\Omega$.

In short, we established in (36) that $\mathcal{H}(\cdot)$ serves as a *Lyapunov function* for the continuous-time system (MF). Since $\mathcal{H}(\cdot)$ is *strictly* concave, the only ICT set is the singleton $\{p_T^*\}$, where $p_T^*$ represents the uniform (and hence entropy-maximizing) measure on $\Omega$ (Benaïm, 2006, Proposition. 6.4).

**Algorithm 1 generates an APT.** Let $(p_T^k)_{k \in \mathbb{N}}$ be the sequence of measures on $\Omega$ generated by Algorithm 1 with the oracle LINEARFINETUNINGSOLVER, and recall that its dual variables are given by $(h^k := \delta \mathcal{H}(p_T^{\pi_k}))_{k \in \mathbb{N}}$. Also, recall the corresponding continuous-time interpolation (Int).

Assumption 7.1 ensures that each dual variable $h^k$ is a well-defined function on $\Omega$ after some iteration $j$, while Assumption 7.2 guarantees the precompactness of $h(\cdot)$. Furthermore, under Assumption 7.3, standard arguments (see, e.g., **Proposition 4.1** of (Benaïm, 2006) or (Karimi et al., 2024)) establish that $h(\cdot)$ generates an APT of the continuous-time flow defined by (MF). Finally, Theorem C.1 ensures that $\{h^k\}_{k \in \mathbb{N}}$ converges almost surely to an ICT set of (MF), which we have already shown to contain only $\{p_T^*\}$.

Therefore, applying the theory of (Hsieh et al., 2021; Karimi et al., 2024), we conclude that, almost surely,

$$\lim_{k \to \infty} h^k = \lim_{k \to \infty} \delta \mathcal{H}(p_T^k) = \lim_{k \to \infty} -\log p_T^k = \delta \mathcal{H}(p_T^*) \qquad \text{in } L_\infty. \tag{37}$$

Since $\Omega$ is compact, (37) implies that, for any smooth test function $\psi$ on $\Omega$, $\langle p_T^k, \psi \rangle \to \langle p_T^*, \psi \rangle$, which completes the proof. $\qquad \square$

# D. Detailed Example of Algorithm Implementation

## D.1. Pseudocode for implementation of Eq. (9)

For the sake of completeness, in the following we present the pseudocode for a possible implementation of a LINEARFINE-TUNINGSOLVER via a first-order optimization method, used to solve (9), as well as within S-MEME. In particular, we present the same implementation we use in Sec. 8, based on Adjoint Matching (Domingo-Enrich et al., 2024), which captures the linear fine-tuning via a stochastic optimal control problem and solves it via regression.

In the following, we adopt the notation from the Adjoint Matching paper (Domingo-Enrich et al., 2024, Apx E.4). We denote the pre-trained noise predictor by $\epsilon^{pre}$, the fine-tuned one as $\epsilon^{\text{finetuned}}$, and with $\bar{\alpha}$ the cumulative noise schedule, as used by Ho et al. (2020). The complete algorithm is presented in Algorithm 2. First, notice that given a noise predictor $\epsilon$ (as Defined in Sec. 2) and a cumulative noise schedule $\bar{\alpha}$, one can define the score $s$ as follows (Song & Ermon, 2019):

$$s(x, t) := -\frac{\epsilon(x, t)}{\sqrt{1 - \bar{\alpha}_t}} \tag{38}$$

---

**Algorithm 2** LINEARFINETUNINGSOLVER (Implementation based on Adjoint Matching (Domingo-Enrich et al., 2024))

---

**input** $N$ : number of iterations, $\epsilon^{pre}$ : pre-trained noise predictor, $\alpha$ regularization coefficient, $m$ : trajectories batch size, $\nabla f$: reward function gradient

1: **Init:** $\epsilon^{\text{finetuned}} := \epsilon^{pre}$ with parameter $\theta$

2: **for** $n = 0, 2, \ldots, N - 1$ **do**

3:     Sample $m$ trajectories $\{X_t\}_{t=1}^{T}$ according to DDPM (Song et al., 2020), e.g., sample $\epsilon_t \sim \mathcal{N}(0, I)$, $X_0 \sim \mathcal{N}(0, I)$

$$X_{t+1} = \sqrt{\frac{\bar{\alpha}_{t+1}}{\bar{\alpha}_t}} \left( X_t - \frac{1 - \frac{\bar{\alpha}_t}{\bar{\alpha}_{t+1}}}{\sqrt{1 - \bar{\alpha}_t}} \epsilon^{\text{finetuned}}(X_t, t) \right) + \sqrt{\frac{1 - \bar{\alpha}_{t+1}}{1 - \bar{\alpha}_t} \left( 1 - \frac{\bar{\alpha}_t}{\bar{\alpha}_{t+1}} \right)} \epsilon_t$$

Use reward gradient:

$$\tilde{a}_T = \nabla f(X_T)$$

For each trajectory, solve the lean adjoint ODE, see (Domingo-Enrich et al., 2024, Eq. 38-39), from $t = T$ to 0:

$$\bar{a}_k = \bar{a}_{t+1} + \bar{a}_{t+1}^\top \nabla_{X_t} \left( \sqrt{\frac{\bar{\alpha}_{t+1}}{\bar{\alpha}_t}} \left( X_t - \frac{1 - \frac{\bar{\alpha}_t}{\bar{\alpha}_{t+1}}}{\sqrt{1 - \bar{\alpha}_t}} \epsilon^{\text{pre}}(X_t, t) \right) - X_t \right)$$

Where $X_t$ and $\tilde{a}_t$ are computed without gradients, i.e., $X_t = \texttt{stopgrad}(X_t), \tilde{a}_t = \texttt{stopgrad}(\tilde{a}_t)$. For each trajectory compute the Adjoint Matching objective (Domingo-Enrich et al., 2024, Eq. 37):

$$\mathcal{L}(\theta) = \sum_{t=0}^{T-1} \left\| \sqrt{\frac{\bar{\alpha}_{t+1}}{\bar{\alpha}_t (1 - \bar{\alpha}_{t+1})}} \left( 1 - \frac{\bar{\alpha}_t}{\bar{\alpha}_{t+1}} \right) \left( \epsilon^{\text{finetuned}}(X_t, t) - \epsilon^{\text{pre}}(X_t, t) \right) - \sqrt{\frac{1 - \bar{\alpha}_{t+1}}{1 - \bar{\alpha}_t}} \left( 1 - \frac{\bar{\alpha}_t}{\bar{\alpha}_{t+1}} \right) \bar{a}_t \right\|^2$$

Compute the gradient $\nabla_\theta \mathcal{L}(\theta)$ and update $\theta$.

4: **end for**

**output** Fine-tuned noise predictor $\epsilon^{\text{finetuned}}$

---

# E. Experiment Details

In this section we provide further details on the experiments.

**Illustrative setting.** Pre-training was performed by standard denoising score-matching and uniform samples, namely $10K$, from the two distributions in Fig. 3a. For fine-tuning, in this experiment we ran S-MEME for 6000 gradient steps in total, for $K = 1, 2, 3, 4$. Notably, k=1 amounts to having a fixed reward for fine-tuning, $-\log p_T^{pre}(x)$. Each round of S-MEME performs $6000/K$ for the particular experiment. In this way we observe the effect of having more rounds of the mirror descent scheme, with same number of gradient updates. Since we utilize Adjoint Matching (Domingo-Enrich et al., 2024) for the linear solver in Algorithm 1, we perform an iteration of Algorithm 2 by first sampling 20 trajectories via DDPM of length 400 that are used for solving the lean adjoint ODE with the reward $-\lambda\nabla\log p_T(x)$ and $\lambda = 0.1$. Subsequently we perform 2 stochastic gradient steps by the Adam optimizer with batch size 2048, initialized with learning rate $4 \times 10^{-4}$. For the density plots in Figure 3 we sampled 80000 points with 100 DDPM steps. To obtain Figure 3d, we computed a Monte-Carlo estimate of $\mathcal{H}(p_T^{\pi_k})$ with an approximation of $\log p_T^{\pi_k}(x)$ resulting from the instantaneous change of variables and divergence flow equation,

$$\log p(x) = \log p_0(x) + \int_0^T \nabla \cdot f(x_t, t)dt, \tag{39}$$

where $f$ is the velocity of the probability-flow ODE for the variance-preserving forward process of the diffusion.

**Text-to-image architecture design.** For obtaining Figure 4, similarly as in the illustrative example we used Algorithm 2 as the linear solver. At each iteration of Algorithm 2 we sample 4 trajectories of length 60 by DDPM, conditioned on the prompt on top of which we perform 10 Adam steps with initial learning rate $3 \times 10^{-7}$ and batch size 8. Each iteration of Algorithm 1 entails 20 iterations of Algorithm 2. We ran Algorithm 1 for $K = 3$. For this experiment, we used $\lambda = 0.1$.

**Text-to-image evaluation.** Evaluating the entropy of $p_T^{\pi_k}$ is computationally prohibitive for the case of the high-dimensional latent of SD-1.5. Consequently, we opted for proxy metrics to quantify how much does the distribution change with increase of $\pi_k$, the FID score for distributional distance and CLIP score for semantic alignment. In addition, we computed the cross-entropy in Table 1 between the Gaussians in the Inception-v3 feature space, where the Gaussians were fitted the same way as in computing the FID score. The FID score, cross-entropy and CLIP score have been computed on 3000 samples from respective conditional distributions.

# F. Diversity Measures

Picking a proper diversity measure for the space of images is non-trivial. For completeness, we provide here additional results for the Vendi score (Friedman & Dieng, 2022), which is a kernel diversity metric. This however again comes with a particular choice of kernel and feature map $\phi$. We picked the standard RBF kernel,

$$\mathbf{k}(x, x') = \exp(-\gamma\|\phi(x) - \phi(x')\|), \tag{40}$$

where we set $\gamma = 50$ and pick the Inception-v3 model feature space. Given computation constraints, we re-ran another experiment with the prompt "A creative architecture." and evaluated the sampled for each MD iteration with 300 images (samples are given in Figure 5) with $\lambda = 0.1$ and initial learning rate for Adam being $10^{-6}$ with 2 MD steps and 100 gradient steps per MD iteration. The reported numbers for Vendi and CLIP scores in Table 2 are computed over 3 seeds.

|  | $\mathbf{p_T^{pre}}$ | **S-MEME 1** | **S-MEME 2** |
|---|---|---|---|
| **Vendi** | 1.53 | $1.7 \pm 0.08$ | $1.63 \pm 0.07$ |
| **CLIP** | 0.22 | $0.22 \pm 0.01$ | $0.22 \pm 0.01$ |

Table 2: FID, CLIP and cross-entropy evaluation of $p_T^{pre}$ and $p_T^{\pi_k}$. For $k = 1, 2$, S-MEME achieves larger distance to $p_T^{pre}$ while preserving high CLIP score.

# G. Additional Text-to-Image Results

In the following, we present additional experimental results obtained via the same text-to-image pre-trained diffusion model introduced in Sec. 8, and with experimental details as presented within Sec. E.

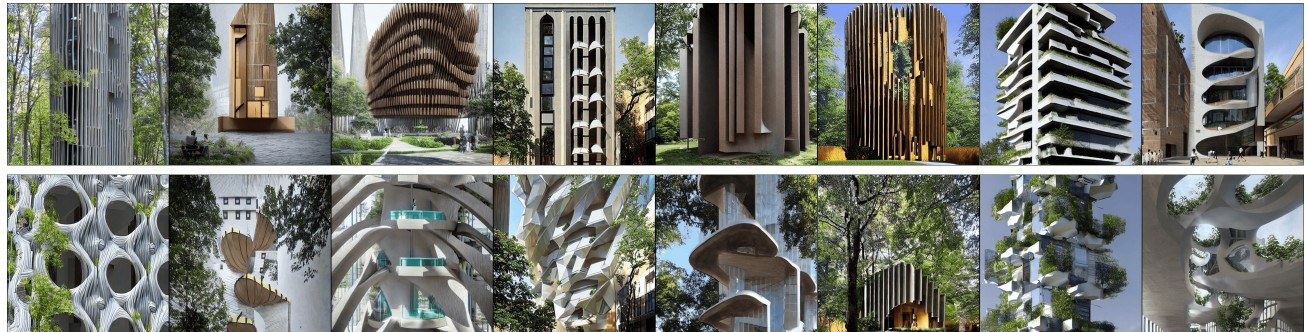

Figure 5: Generated images from $\pi^{pre}$ (top) and $\pi_3$ (bottom) for a fixed set of initial noisy samples using the prompt "A creative architecture.".

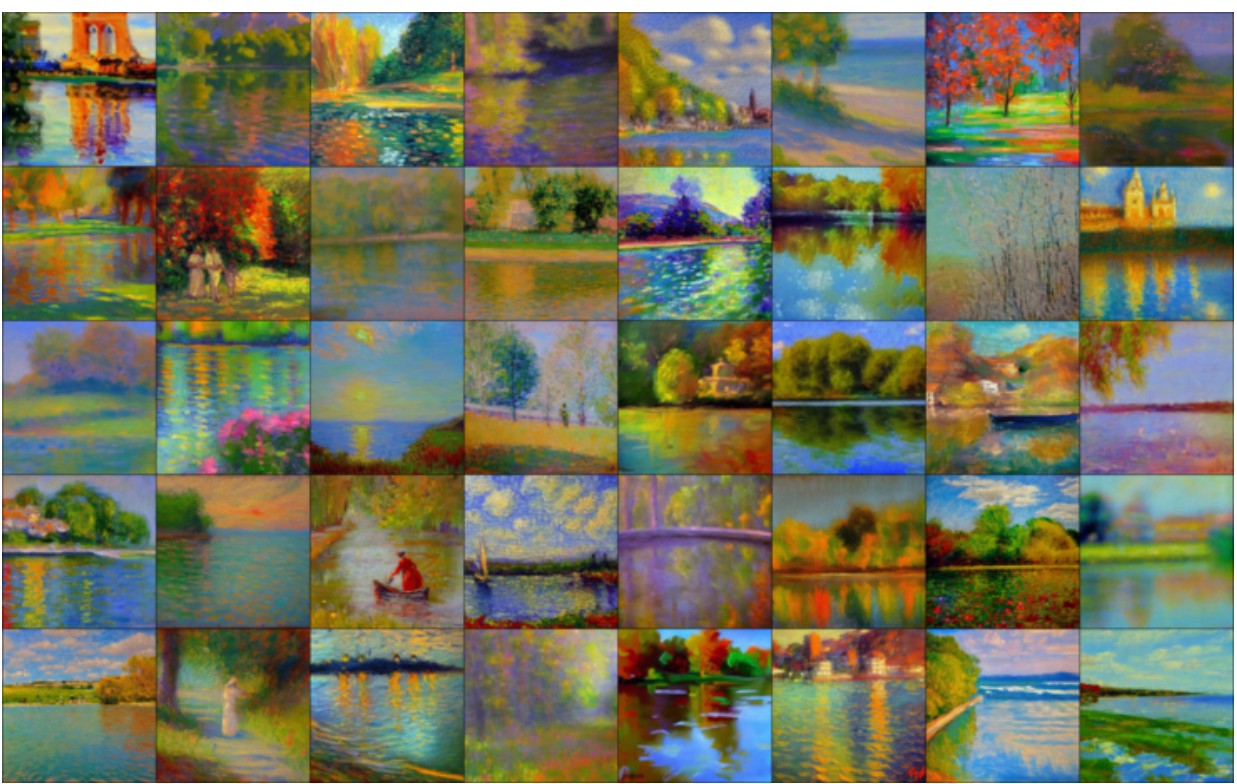

Figure 6: Generated images from $\pi^{pre}$ with prompt "A creative impressionist painting."

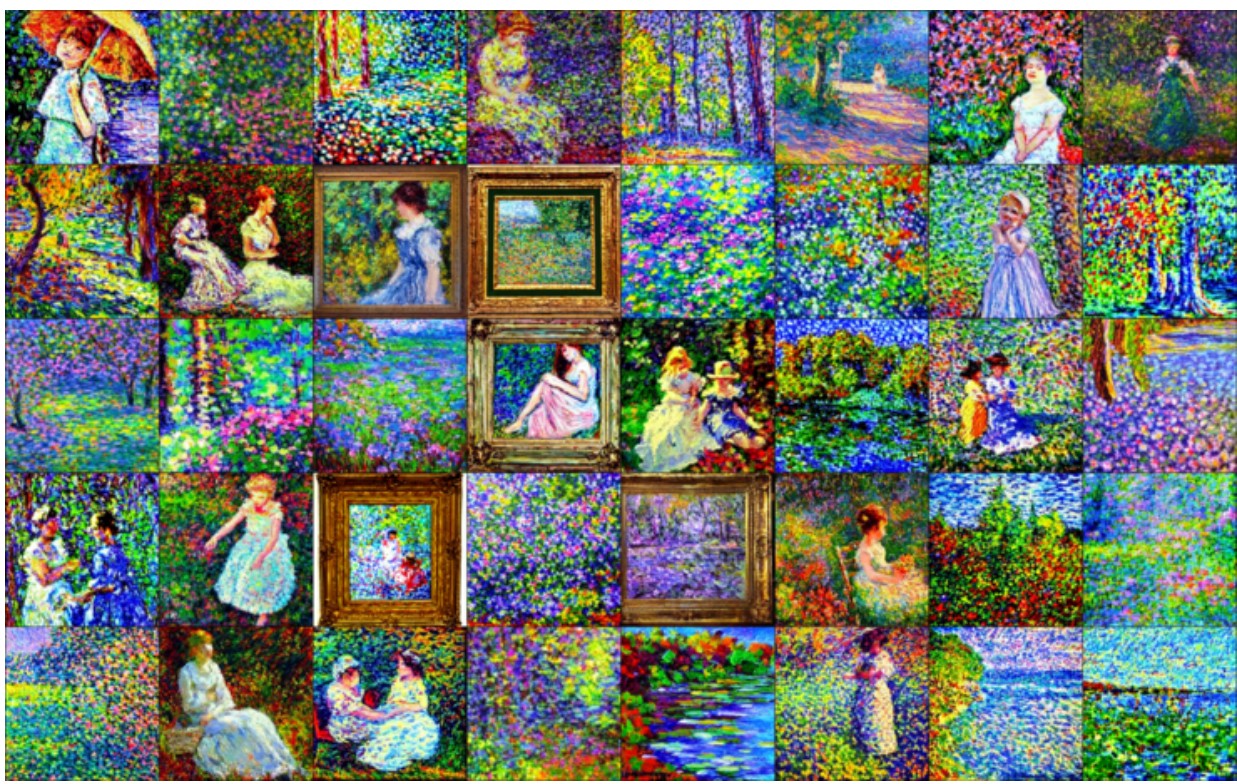

Figure 7: Generated images obtained via fine-tuning of $\pi^{pre}$ via S-MEME with prompt "A creative impressionist painting."

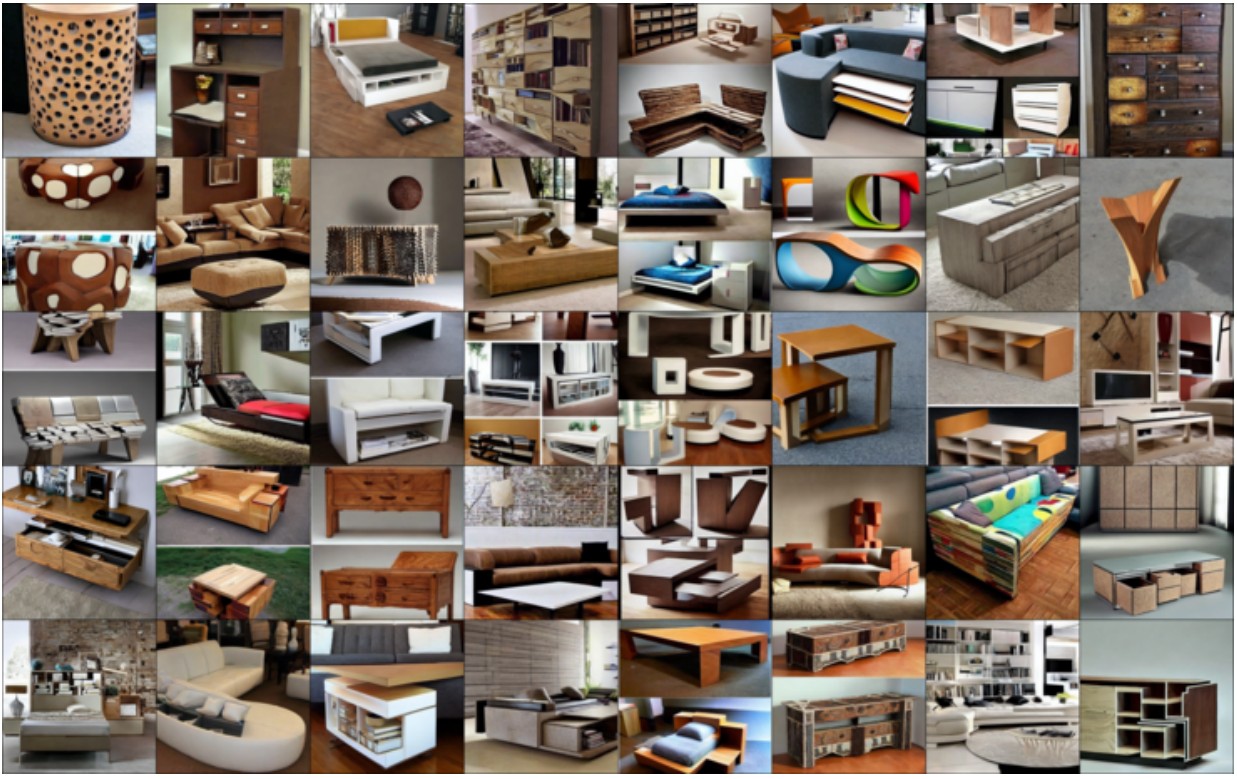

Figure 8: Generated images from $\pi^{pre}$ with prompt "Creative furniture."

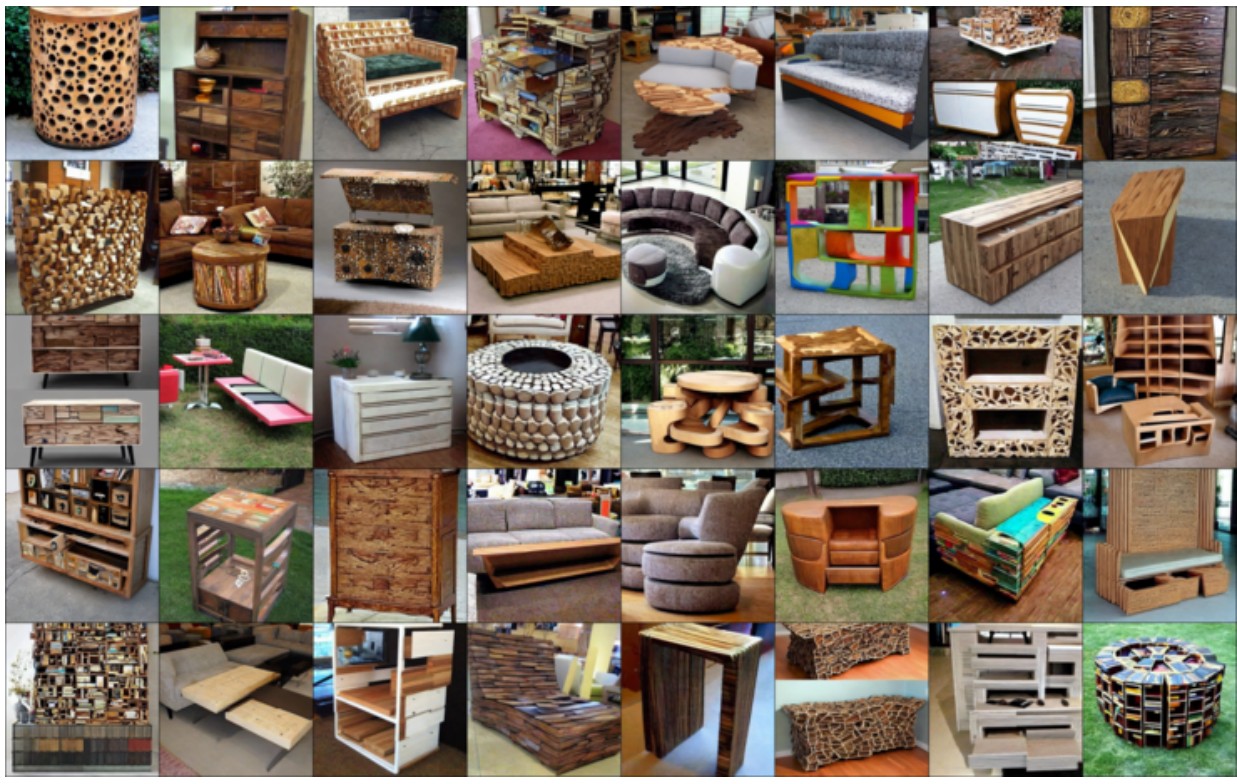

Figure 9: Generated images obtained via fine-tuning of $\pi^{pre}$ via S-MEME with prompt "Creative furniture."

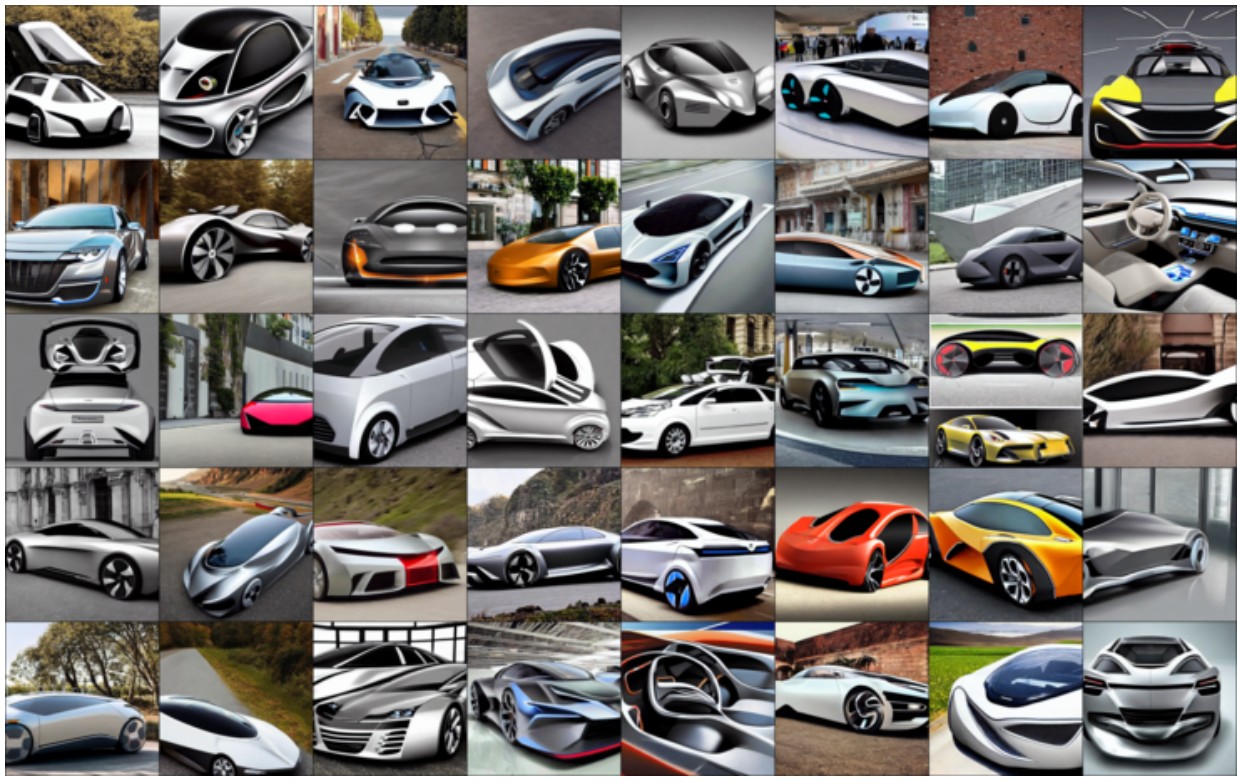

Figure 10: Generated images from $\pi^{pre}$ with prompt "An innovative car design."

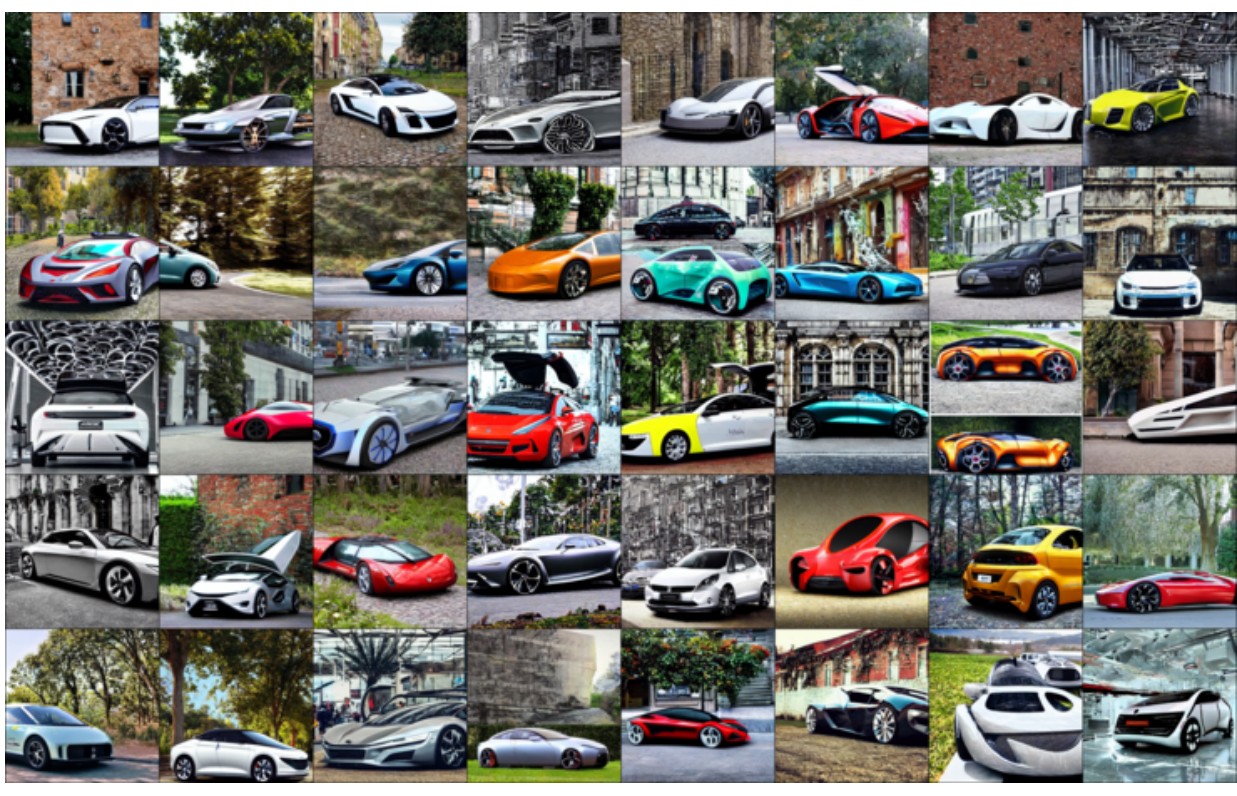

Figure 11: Generated images obtained via fine-tuning of $\pi^{pre}$ via S-MEME with prompt "An innovative car design."

