# OpenReview forum: "Provable Maximum Entropy Manifold Exploration via Diffusion Models"
_ICML.cc/2025/Conference — ICML 2025 poster_

### Official Review · Reviewer_FUKh · 2025-03-07

**Overall Recommendation:** 3

**Summary:**

The authors consider the problem of exploration in planning and decision-making problems.  This problem has many applications including to the exploration-exploitation paradigm in reinfocement learning.  While in most applications the exploration step is performed by sampling from a Gaussian process, the authors consider more general exploration distributions modeled by neural networks.  Specifically, for datasets which lie near lower-dimensional manifolds, the authors aim to sample from entropy-maximizing distributions, which maximize entropy aver the lower-dimension manifold.

 The main challenge encountered by the authors is that while generative models are able to generate points, the authors wish to explore “atypical” subsets of this distribution which may have convenient properties for the given application.  In order to compute this entropy-maximizing distribution and sample from it, the authors use the fact that the score function in diffusion models is (in the limit as t —> 0) given by log density of the distribution from which the data was sampled.  This allows the authors to explore the space of distributions in an entropy-maximizing manner.  Next, the authors apply their exploration method to design an algorithm based on mirror descent for the exploring the manifold by sampling from a maximum-entropy distribution on the manifold.

## update after rebuttal

Thank you for the helpful clarifications.

**Claims And Evidence:**

The authors provide theoretical guarantees which show (i) that their entropy maximizing exploration step is optimal and (ii) that their algorithm (Algorithm 1) converges to the maximum entropy density after infinitely many iterations of their algorithm. Visually, the designs from their method appear (subjectively) to be more unusual/less conventional than the baseline model, but possibly at the cost of being less realistic and of lower quality.  Empirically, the authors implement their algorithm on an architecture dataset where the goal is to explore atypical architectural designs.

**Essential References Not Discussed:**

N/A

**Experimental Designs Or Analyses:**

Please see "Methods And Evaluation Criteria" question above.

**Methods And Evaluation Criteria:**

The theoretical results prove the optimality of the authors exploration method (Theorem 5.2), and the "asymptotic correctness" of their algorithm (Theorem 7.1). However, they do not provide any non-asymptotic guarantees for their algorithm.

For empirical criteria, the architecture dataset used by the authors provides a good illustration of the exploration problem, and allows the authors to compare their algorithm to baseline methods on a compelling application.   Visually, the designs from their method appear (subjectively) to be more unusual/less conventional than images sampled without any iterations of their mirror descent algorithm distribution (which serves as a baseline model), but possibly at the cost of being less realistic and of lower quality.  Thus, it would be good to discuss the apparent tradeoff in visual sample quality.

**Other Comments Or Suggestions:**

Please see "Other Strengths And Weaknesses" question above.

**Other Strengths And Weaknesses:**

The main strengths of the paper are in proposing a method for entropy-maximizing exploration, and for providing an algorithm with one-step optimality and (asymptotic) convergence guarantees for this problem.  The main weaknesses are (i) that the convergence guarantees are asymptotic (there is no guarantee on how fast their method converges) and (ii) the empirical results could be made a bit more clear.  Specifically I found it a bit unclear what are the baseline model(s) being compared to empirically, and what the tradeoff is between image quality and "creativity" or "entropy" when the authors.  If I understand correctly, from Figure 3 there appears to be a tradeoff, so it would be good for the authors to discuss this in more detail.

**Questions For Authors:**

Can the authors clarify in the empirical section which baseline model(s) they are comparing to?  I believe the baseline model is the stable diffusion model, but this is a bit unclear to me.    Also, if I understand correctly, from Figure 3 there appears to be a tradeoff, so it would be good for the authors to discuss this in more detail.

**Relation To Broader Scientific Literature:**

Empirically, the authors compare to the stable diffusion model as a baseline.  However, it may be good to include additional baselines to compare to, if this is feasible.  Also, it is a bit unclear from the captions in Figure 3 and Table 1 which models are the authors' and which are the baselines (I believe that $\pi_pre$ is the authors and $\pi_3$ is a baseline model, but this could be made a bit more clear)

**Theoretical Claims:**

I did not carefully check the proofs in the appendix.

---

> ### Author Rebuttal · Authors · 2025-04-01
>
> We thank the Reviewer for appreciating our work and asking interesting questions. In the following, we address several important points mentioned within the review that can hopefully let the Reviewer appreciate more the content of this work.
>
> **Asymptotic convergence guarantees**
> We thank the Reviewer for raising these concerns and would like to provide a clarification: We can in fact establish convergence guarantees under three different assumptions, ordered by increasing generality:
> - Perfect fine-tuning: If fine-tuning is exact, our algorithm terminates in a single step (see Section 5).
> - Unbiased noise oracle: When the noise oracle is unbiased (i.e., $ b_k = 0$ in Eq. (16)), standard mirror descent analysis yields a convergence rate of $\mathcal{O}(k^{-1/2})$ (see [3]).
> - General bias term: If the noise oracle in Eq. (16) includes a general bias term, then under the arbitrary slow decay assumption in Eq. (18), a polynomial-time guarantee is no longer feasible [3]. In this case, stochastic approximation techniques are required, and the best achievable rate is  $\tilde{\mathcal{O}}((\log\log k)^{-1})$, which follows from our proof. We chose to present an asymptotic result rather than explicitly stating this rate, as the difference is negligible and in this case it is conventional to present the asymptotic result [3,4].
>
> Among these, the third setting is the most practical, which is why we focused on it in Section 7. We will incorporate these clarifications into the revision.
>
>
> **Empirical results clarification**
> Within the Experimental Evaluation section, we evaluate qualitatively (i.e., visually) and quantitatively (i.e., via the metrics within Fig. 2-d and Table 1) the performance of models $\pi_k$ obtained by fine-tuning a pre-trained model $\pi^{pre}$ for $k$ iterations of S-MEME. In the case of text-to-image experiments, Fig. 3 shows on the top row a set of images obtained by sampling the pre-trained model $\pi^{pre}$, which in this case corresponds to Stable Diffusion 1.5 [1] trained on the LAION-5B dataset [2], while the bottom row shows images sampled via $\pi_3$, which is the diffusion model obtained after $3$ iterations of our algorithm. Qualitatively, one can visually notice that the images within the top row appear mostly gray and similar to each other, while the images from the bottom row show more diversity both among them and compared with the ones above, while preserving semantic meaning. Quantitatively, ideally we would want to estimate the entropy of the marginal density induced by the fine-tuned model, but this is hard in practice, as explained within Sec. 8.2. As a consequence, within Table 1 we show several proxy metrics to give numerical insights about the performances of the fine-tuned models. Within this table, the label 'S-MEME 1' refers to the model obtained after one single iteration of S-MEME, 'S-MEME 2' after two iterations, and 'S-MEME 3' after three iterations. Crucially, we show that the FID and cross-entropy scores, which aim to capture the degree of diversity of the marginal density induced by a model $\pi_k$ from the pre-trained model, increases over the iterations of S-MEME, meanwhile the CLIP score, which assesses the naturalness, or image quality, of the generated samples, is kept high across increasing iterations of S-MEME. Clearly, the trade-off between surprise maximization (i.e. diversity from pre-trained model) and naturalness due to regularization with the pre-trained model, can be chosen arbitrarily by tuning the $\{\alpha_k\}$ parameters of the algorithm, which manage this trade-off as shown In Eq. (9). The experimental results in Sec. 8 aim to evaluate a specific parameter choice to show practical relevance of the proposed scheme.
>
> We thank the Reviewer for these question sand hope that the explanations given can help the Reviewer better appreciate our work. We will update a revised version of the paper with better clarification of these aspects.
>
> **References**
>
> [1] Rombach et al., High-resolution image synthesis with latent diffusion models. CVPR 2022.
>
> [2] Schuhmann et al., An open large-scale dataset for training next generation image-text models. NeurIPS 2022.
>
> [3] Karimi et al., Sinkhorn flow as mirror flow: A continuous-time framework for generalizing the sinkhorn algorithm. AISTATS 2024.
>
> [4] Borkar et al., The ODE method for convergence of stochastic approximation and reinforcement learning. SIAM Journal on Control and Optimization 2000.

---

### Official Review · Reviewer_kBVm · 2025-03-13

**Overall Recommendation:** 2

**Summary:**

This paper introduces a maximum entropy manifold exploration problem. They proposes a modification to the pretrained diffusion model to maximize an entropy objective function. They also proposed an algorithm to solve this optimization problem. They supported their results with numerical experiments.

**Claims And Evidence:**

Yes.

**Essential References Not Discussed:**

Essential references are discussed.

**Experimental Designs Or Analyses:**

I didn't check the code.

**Methods And Evaluation Criteria:**

Their problem is new, so I feel this point is not applicable here...

**Other Comments Or Suggestions:**

1. Isn't the maximizer of the entropy function just the uniform distribution over $\Omega^{pre}$?
2. How related are the two objectives (7) and (9)?
3. I feel Section 4.3 is not presented very clearly: What is the motivation for using $s^{pre}$ in the actual implementation instead of $s^{\pi}$?
4. I am not very familiar with RL, but I feel in general we not only want to do exploration but also want to do exploitation and maximize a reward function. This is also the case for diffusion guidance: practioners want to guide the sample generation towards a direction that maximizes certain reward function. Can the authors discuss extension in this aspect?

**Other Strengths And Weaknesses:**

The problem is novel, while the motivation is not that clear to me

**Questions For Authors:**

1. Isn't the maximizer of the entropy function just the uniform distribution over $\Omega^{pre}$?
2. How related are the two objectives (7) and (9)?
3. I feel Section 4.3 is not presented very clearly: What is the motivation for using $s^{pre}$ in the actual implementation instead of $s^{\pi}$?
4. I am not very familiar with RL, but I feel in general we not only want to do exploration but also want to do exploitation and maximize a reward function. This is also the case for diffusion guidance: practioners want to guide the sample generation towards a direction that maximizes certain reward function. Can the authors discuss extension in this aspect?

**Relation To Broader Scientific Literature:**

It is related to diffusion guidance, an important technique used in the diffusion model commmunity. The paper also mentioned applications in molecular generation.

**Theoretical Claims:**

No.

---

> ### Author Rebuttal · Authors · 2025-04-01
>
> We thank the Reviewer for reading our work. In the following, we address several fundamental points and questions mentioned within the review that can hopefully let the Reviewer appreciate more the content of this work.
>
> **Motivation of the work**
> In the following, we aim to make clear the main motivation of this work. Pre-trained generative models can generate plausible objects of a certain data type, e.g. valid images or molecules. Nonetheless, the probability of sampling rare, novel, objects will be very low. This is because the model has been trained on a dataset of existing objects to approximate the data distribution. Therefore rare objects in the data will be rare to sample. In this work, we adapt the generative model so that its induced distribution is not proportional to that of existing data, but is shifted towards lower-probability regions where novel designs can be sampled, while preserving plausibility (i.e., staying within the approximate data manifold). To the best of our knowledge, this is a fundamental (open) problem for discovery via generative models.
>
> **Question 1**
> Yes, it is. In fact, if $\Omega_{pre}$ would be a known set representable in a computer one could simply define a uniform distribution on such a set and then sample from it. But what if $\Omega_{pre}$ is an unknown and possibly very complex set, e.g., space of valid molecules or images, that cannot be represented in a computer if not implicitly via a generative model? How can we explore such a set? This work tackles exactly this problem, which is both fundamental and non-trivial. We thank the Reviewer for asking this question as we believe it is of central importance, and we will make sure to further clarify this point in a revised version of the work.
>
> **Question 2**
> Equation (7) describes an optimization problem over the set of densities supported over a set $\Omega_{pre}$ represented only implicitly by a pre-trained diffusion model. This fully captures the manifold exploration problem. On the other hand, Eq. (9) is an unconstrained KL-regularized optimization problem of a reward function obtained by linearizing the entropy, as explained in details within Sec. 4. As explained in Sec. 6, a concave function like entropy can be maximized via a Mirror Descent scheme [1], which relies on a sequence of simpler optimization problems. In this context, Eq. (7) represents the concave optimization problem, while Eq. (9) captures each simpler optimization problem. Crucially, via this viewpoint, we can use scalable methods to solve each smaller problem to solve the complex exploration problem. We will make sure to make this more explicit within an updated version of the work.
>
> **Question 3**
> Within this work, $s^{pre}$ is the score of a pre-trained diffusion model that one wishes to fine-tune by optimizing a regularized measure of surprise (Eq. (9)).This quantity has to be estimated with respect to the previous model, which at the first iteration of S-MEME corresponds to the pre-trained model with score $s^{pre}$. In practice, as mentioned within Sec. 4.3, the score of the fine-tuned model $s^\pi$ can be initialized as $s^\pi = s^{pre}$. After this step, there is no difference in using $s^\pi$ or $s^{pre}$ to estimate the induced marginal density as these score networks are equal. We will make sure to make this more clear in order to prevent any doubt.
>
> **Question 4**
> Reinforcement learning spans several problems, including pure-exploration ones, e.g., [2,3], where the goal is to explore a certain space, typically via maximization of an intrinsic reward, which in the case of this work would correspond to surpise (i.e., the entropy first variation) in Eq. (9) and (10). Pure exploration problems often are relevant on their own. In this case, surprise maximization is clearly relevant for discovery applications where it can make it possible to sample surprising designs, or, as mentioned by Reviewer WBXm, to de-bias a given pre-trained model by inducing a more balanced distribution than the data distribution. Nonetheless, ideas from pure exploration can be used for exploration schemes in exploration-exploitation settings where there is an unknown quantity (e.g., a reward function) to be learned and optimized. In this case, the exploration principle introduced within Eq. (9), which makes it possible to scalably maximize a measure of surprise, could be used as a way to regularize an exploratory scheme for reward-learning in black-box optimization settings, e.g., [4].
>
> **References**
>
> [1] Nemirovski et al., Problem complexity and method efficiency in optimization, 1983.
>
> [2] Hazan et al., Provably efficient maximum entropy exploration. ICML 2019.
>
> [3] Mutny et al., Active exploration via experiment design in markov chains. AISTATS 2023.
>
> [4] Uehara et al., Feedback Efficient Online Fine-Tuning of Diffusion Models. ICML 2024.

---

### Official Review · Reviewer_Q9Dh · 2025-03-13

**Overall Recommendation:** 3

**Summary:**

The paper presents a framework to perform optimal exploration of the data manifold defined by a pre-trained diffusion model. This can be useful whenever one wants to sample using diffusion models and explore the full data region within the learned data manifold. The approach that the paper proposes is based on self-guided exploration using the density estimated by the diffusion model itself. Using a connection between the entropy of the density learned by the diffusion model and its score function, they propose a sequential fine-tuning procedure that results in a fine-tuned version that should lead to optimal exploration. The fine-tuning procedure is derived using the connection of diffusion model and reinforcement learning and by using mirror descent theory. They evaluate the proposed method on a toy dataset and an image-to-text task.

**Claims And Evidence:**

The theoretical claims done in the paper seem supported by proofs. Claims regarding the scalability and effectiveness of the proposed method are supported by two (a toy and a text-to-image) experiments. (More on the experiments below).
I have just one minor comment, and I  might be wrong here, so I ask the authors to correct me if I am wrong. There is a constant repetition that the method proposed in the paper does not rely on explicit uncertainty quantification. I agree it's not explicit, but as the method is framed in terms of density estimation it strongly relies on a sort of uncertainty of the score function it seems. Because the constant repetition seems to give the message that the method wants to be completely detached from the concept of uncertainty in any of the forms, which might be a bit misleading.

**Essential References Not Discussed:**

-

**Experimental Designs Or Analyses:**

I have checked the experimental details provided in the appendix. As mentioned above, there are not so many details on the training set and how the score was trained in the first experiment. It would be also interesting to know how expensive it is to perform the fine-tuning procedure the authors describe in algorithm 1 as it seems to consist of a nested loop. Also how did you usually choose the number of $K$ refinement iterations and the number of $N$ iterations in the inner loop? Additionally, in line 844 you mention 4 trajectories and then you mention a batch size of 8. Are these two related?

**Methods And Evaluation Criteria:**

As mentioned above, to show the effectiveness of the proposed framework, they consider two different experiments: a 2D-toy experiment and a text-to-image experiment involving stable diffusion.

- The first toy experiment is a nice and intuitive way to present the problem the paper wants to tackle and what are the results of the proposed method. It would be helpful to have all the plots with the same x-range and maybe a different `cmap` as in (b) some samples are rarely visible. I think it would also be curious to have a plot of the true density and the samples used to train the pre-trained model. Also, there are no details on how the authors trained the diffusion model in the appendix for that specific example.
- Regarding the second experiment: the authors consider a pre-trained text-to-image diffusion based on based on stable diffusion and perform experiments for two different prompts. They measure results in terms of FID, CLIP, and distance between the two marginal distributions. You mention that you are showing samples for every iteration of the fine-tuning in Figure 9, but it is not clear if that is the case. Also, while the evaluation makes sense, finding a task where the exploration is needed is difficult and I am not fully convinced that the image domain is the best application. Also by comparing Fig. 4 and Fig.5 it is difficult to understand that the fine-tuned model is exploring the manifold more as all the samples look similar, and none of them resemble the one of the pre-trained model. Those results might be similar to getting samples by using two different guidance strengths, but I might be wrong. Maybe having an additional experiment on molecules with specific properties (as the authors mentioned in the paper) would make the paper stronger.

**Other Comments Or Suggestions:**

- Line 831 Wince instead of Since
- There might be a possible mistake in the pseudocode in the appendix I guess. In the adjoint ODE, there is a $k$ subscript that it's not clear where it comes from (if it comes from the outer loop then you might have to include it as input). Also if the initial $t=T$, then the first $\bar{a}$ considered is $T+1$ which does not exist.

**Other Strengths And Weaknesses:**

-

**Questions For Authors:**

- Where does the $\eta \in [0,1]$ come from in the reverse process in Eq. 2? It's not justified and it does not appear in the forward process. Can the authors comment on this?
- Why in proposition 1 the noise distribution $p_0$ is a truncated Gaussian? What's the bounding interval in this case? For standard diffusion models, the noise distribution at time T and not 0 approximately converges to a standard Gaussian for variance-preserving processes. Also in the section "Score matching and generation" the author is using $p_0$ for the data distribution. Indeed, at some point the author starts describing everything in terms of the time going backward, and when $t=T$ we get that the backward marginal $p_T$ corresponds to the data distribution. Therefore, the notation used in the paper can be improved to make the paper more clear.
- It seems that the algorithm depends at each iteration on a different ${\alpha_k}_1^K$ that weighs exploration and exploitation. How should one decide the values of ${\alpha_k}_1^K$?
- I might have a naive question: is Assumption 7.1 always satisfied? I understand that if we assume that both models have support on $\mathbb{R}^d$ this is true, but can it happen that by maybe forcing the model to explore too much, due to approximations and noise, the algorithm will end up sampling from regions of the space where there were no data initially and were the pre-trained model was not able to sample from? Similar to the case of using a too strong guidance strength? Like in Fig. 2 it seems that the tuned model gets samples on the true data support but where it seems that the pre-trained model gets no samples.

**Relation To Broader Scientific Literature:**

They present connections to broader scientific literature in the related works section, which is nicely written and very detailed.

**Theoretical Claims:**

I checked the theoretical claims the authors have in the paper, but I cannot guarantee that the proofs are completely correct.

---

> ### Author Rebuttal · Authors · 2025-04-01
>
> We thank the Reviewer for the interesting questions. In the following, we address several points that can hopefully let the Reviewer appreciate more this paper, and which will be included in a revised version.
>
> **Uncertainty quantification**
> We agree, our method does not employ *explicit* uncertainty quantification, but arguably relies on a form of *implicit* uncertainty quantification, although not the classic Bayesian one.
>
> **Toy experiment**
> - The y-range of plot 2.a should be the same as plots 2.b and 2.c.
> - Since Fig. 2.b and 2.c are histogram plots, for a smooth cmap, points with low probability have a similar color to points with zero probability. We will try to overcome this issue by testing non-smooth cmaps.
> - The original density is in Fig. 2.a, namely two uniform distributions.
> - Pre-training was performed by standard denoising score-matching and uniform samples (namely 10K) from the two distributions in Fig. 2.a.
>
>
> **Text-to-Image experiment**
> - The references to Fig. 9 in Sec. 8 should be references to Fig. 3.
> - We agree with the Reviewer that the method presented in this work is relevant for molecular spaces. Nonetheless, applications in scientific discovery go beyond the scope of this paper, which captures mathematically a novel and relevant problem, proposes fundamental algorithmic advances, and provides a novel type of theoretical analysis for diffusion fine-tuning. We believe that exploration in the visual domain is relevant for multiple applications, and particularly amenable for evaluation as it does not require specific background knowledge.
> - Since the method proposed directly maximizes a gradient of entropy, one can show that for $\alpha \geq 1$ (in Eq. 9), the entropy value increases monotonically. One could attempt at increasing entropy by reducing guidance strength, but this would result in being closer to the unconditional distribution thus changing the support significantly. On the contrary, our algorithm (provably) explores within the support of the conditional distribution via entropy maximization, which is substantially different.
>
> **Computational complexity and hyperparameter selection**
> Under exact fine-tuning one iteration is sufficient (see Sec. 5). In practice, we show in Sec. 8 that very few iterations (e.g., $K=3$) can lead to useful fine-tuned models. Therefore, the computational cost aligns with typical fine-tuning, e.g. [3]. The parameter $N$ is significantly problem dependent, and as in the case of Adjoint Matching [3], it requires experimental tuning.
>
> **Question 1**
> It is common to express the reverse process of a diffusion process via a SDE parametrized by a parameter $\eta \in [0,1]$ (e.g., in [1]). The processes obtainable for any valid $\eta$ induce the same marginal distributions.
>
> **Question 2**
> - The truncation in Prop. 1 is merely a technical aspect to build a theoretical formulation that matches the real sampling process, which always gives finite values. The bounding interval can be any as long as it is finite. Therefore it can fully capture any real scenario.
> - Diffusion notation uses $0$ for the time-step corresponding to data (e.g., [1]), while fine-tuning works denote by $0$ the noise level (see, e.g., [2]). This is formally not fully correct, since to compute closed-form the noise distribution one has to take the limit to infinity. A recent rigorous solution (see, e.g., [1]) is the one used in our paper. Nonetheless, we agree with the Reviewer that the sign flip might create confusion and have already added a drawing that clarifies this in a new version.
>
> **Question 3**
> Sec. 4 shows that given exact fine-tuning, the algorithm convergences in $1$ iteration with $\alpha_1 = 1$. It should be set to lower values to prioritize the surprise term in Eq. 9, and to higher values to prioritize regularization, as discussed in Sec. 4.4. When considering apx. fine-tuning oracles, Theorem 7.1 gives necessary conditions on $\alpha_k = 1/\gamma_k$ for the guarantees to hold. In practice, $\alpha_k$ should be experimentally tuned for the specific application.
>
>
> **Question 4**
> Ass. 7.1 could be difficult to verify in some cases. However, it can be relaxed as $
> supp(p_T^{\pi_k}) \subset \tilde{\Omega} \text{ for all } k, $ and $ supp(p_j^{\pi_k}) = \tilde{\Omega} $ for some $ j.
> $ Using the same analysis, we can show that the algorithm can solve the exploration problem on $\tilde{\Omega}$. In particular, if $\tilde{\Omega}$ approximates the true support, as in Fig. 2, our analysis guarantees exploration of the approximate support.
>
> **References**
>
> [1] Zhao et al., Scores as Actions: a framework of fine-tuning diffusion models by continuous-time reinforcement learning.
>
> [2] Uehara et al., Feedback Efficient Online Fine-Tuning of Diffusion Models. ICML 2024.
>
> [3] Domingo-Enrich et al., Adjoint Matching: Fine-tuning Flow and Diffusion Generative Models with Memoryless Stochastic Optimal Control. ICLR 2025.

---

> > ### Comment · Reviewer_Q9Dh · 2025-04-05
> >
> > I would like to thank the authors for answering my questions. I have adjusted my score accordingly. I still think that having an additional experiments where exploration is really needed will make the paper and the method stronger, therefore I would be super interested in seeing that in the updated version.

---

### Official Review · Reviewer_WBXm · 2025-03-21

**Overall Recommendation:** 5

**Summary:**

This paper considers the problem of exploring the underlying data manifold learned with a diffusion model.

It formulates the problem as maximizing the entropy of the probability distribution, and proposes to solve it with a KL-regularized optimization of the first variation of the entropy. It proves that the optimal is obtained given perfect training optimization.

Furthermore, it considers the fact that in practice the training and optimization have errors, and propose an iterative fine-tuning algorithm. It proves that this algorithm converges to the optimal solution under general and realistic assumptions.

The method is theoretically connected to continuous-time RL.

The method is empirically validated with an illustrative example and fine tuning a Stable Diffusion model for text-to-image task.

## update after rebuttal
Thank you for providing the theoretical results and rationales for the evaluation metrics. I agree with these rationales. I think it would be interesting to explore the application of this method such as drug discovery, as future directions.

**Claims And Evidence:**

The claims made in the submission are supported by clear and convincing evidence.

**Essential References Not Discussed:**

This is not directly comparable, but an interesting line of work that might be related, for the problem of manifold exploration/sample bias reduction: Geometry-Based Data Generation (Lindenbaum, et al., 2018, Neurips), Geometry-Aware Generative Autoencoders for Warped Riemannian Metric Learning and Generative Modeling on Data Manifolds (Sun, et al., 2025, AISTATS). These methods solve the problem from a geometrical perspective.

**Experimental Designs Or Analyses:**

The experimental designs are sound. See **Methods And Evaluation Criteria.**

Besides, for future work, it would showcase the utility of this model if tested on broader contexts, such as molecule generation problems mentioned in the paper.

**Methods And Evaluation Criteria:**

The method makes sense, and is practical.

The toy example is evaluated with entropy, the target, and illustration, which makes sense and is intuitive.

For the text-to-image experiment, the CLIP score shows that the p.d.f. still has the same support, and the FID and cross entropy show it deviates from the pretrained model. For the latter, it is implied that the p.d.f. might have higher entropy because if deviates from the original one with low entropy. I understand that entropy is hard to estimate on this space, but would other diversity metrics such as Inception Score or Vendi score be a better surrogate?

Another interesting experiment would be first fine tune a stable diffusion model on some biased dataset to introduce imbalance, and see if the proposed method can recover it.

**Other Comments Or Suggestions:**

typo in the formula in line 95, right column: $x$ should be $x_t$; also, should be minimizer, not maximizer.

**Other Strengths And Weaknesses:**

## Strengths:

1. The paper is very well written and theoretically solid, with proofs of convergence to the proposed objectives.
2. The paper did not stop at the ideal case of perfect pretraining and optimization, but admitting those cases are not attained in practice, and further proposes a solution for this, making the method more practical. The theoretical assumptions for the method are justified.
3. The method is directly usable on pretrained diffusion models and has potential applications in tasks requiring exploration.

## Weaknesses:

1. The evaluation metric for the image generation task can be improved (see **Methods And Evaluation Criteria)**
2. The method could be showcased on tasks with a stronger motivation for manifold exploration (such as drug discovery). The current example of image generation is not very intuitive.

**Questions For Authors:**

1. Are there empirical or theoretical results on the scalability of this method, especially on the larger pretrained models?
2. In assumption 7.3 and thm 7.1 do we assume inifite steps? k, infinite sum? Then is there an estimation of convergence rate?
3. In terms of connection to RL, is it just notational? It seems that the problem is formulated and solved without relying on properties or algorithms of RL.

**Relation To Broader Scientific Literature:**

The paper deals with the data manifold exploration problem, a common problem in real data.

The paper is related to diffusion models, and can be applied as fine tuning steps to existing models.

As potential future work, the method might be useful for improving the exploration of drug discovery and other generative-discovery tasks.

**Theoretical Claims:**

The theoretical claims make sense to me. I skimmed through the proofs in the appendix and I think they are correct.

---

> ### Author Rebuttal · Authors · 2025-04-01
>
> We thank the Reviewer for recognizing our work as very well written, theoretically solid, and practical. In the following, we address several points and questions mentioned within the review.
>
> **Inception Score (IS) and Vendi Score (VS).**
>
> IS: To the best of our understanding, the Fréchet Inception Distance (FID) has been introduced as a way to mitigate the inability of the IS score to recognize intra-class mode collapse [1], which is related to what we are aiming to measure in our setting. In particular, in our setup with images, the generated points would all be part of one class. As a consequence, IS might not be a better evaluation measure than FID.
>
> VS: We agree with the Reviewer that this measure of diversity might be relevant in this context. Nonetheless, choosing a relevant kernel in practice might be non-trivial, and if the feature space is obtained via a non-linear map, which is typically the case, it is unclear how well this measure aligns with entropy. In any case, we are grateful for this suggestion and will certainly explore it in this context.
>
> **Relevance to de-bias pre-trained generative models**
> Although not strictly related to the motivation presented in this work, we agree that the presented method is very relevant to de-bias generative models, and further research could focus on its potential impact on this important problem.
>
> **Question 1.**
> The presented method can run using any state-of-the-art (linear) diffusion model fine-tuning scheme, such as Adjoint Matching [2] or alternative methods, e.g., [3]. To the best of our knowledge, these methods are very scalable and they have been shown to successfully fine-tune large-scale pre-trained models for images [2,3], molecules [3], and proteins [3], among others. Crucially, due to Eq. (12) within Sec. 4.3, our algorithm becomes effectively as scalable as the state-of-the-art methods mentioned above, which have already proved successful in relevant real-world applications with larger pre-trained models [2,3].
>
> **Question 2.**
> We can establish convergence guarantees under three different assumptions, ordered by increasing generality:
> -  Perfect fine-tuning: If fine-tuning is exact, our algorithm terminates in a single step (see Section 5).
>
> - Unbiased noise oracle: When the noise oracle is unbiased (i.e., $\( b_k = 0 \)$ in (16)), standard mirror descent analysis yields a convergence rate of $\( O(k^{-1/2}) \)$ (see [7]).
>
> - General bias term: If the noise oracle in Eq. (16) includes a general bias term, then under the arbitrary slow decay assumption in Eq. (18), a polynomial-time guarantee is no longer feasible [6]. In this case, stochastic approximation techniques are required, and the best achievable rate is  $\tilde{\mathcal{O}}((\log\log k)^{-1})$, which follows from our proof. We chose to present an asymptotic result rather than explicitly stating this rate, as the difference is negligible and in this case it is conventional to present the asymptotic result [6,7].
>
> Among these, the third setting is the most practical, which is why we focused on it in Section 7. Notably, our results show that convergence does not require running $\( k \to \infty \)$ even with the most general assumptions (i.e., case 3 above).
>
>
> **Question 3.**
> The presented version of S-MEME relies on optimal control methods as discussed in Sec. 4.2. Nonetheless, it is standard to refer to this recent problem of optimization over the space of admissible state distribution to maximize a known functional as Convex RL [4] or General Utilities RL [5]. This is the main reason why we use the RL notation. Moreover, to the best of our understanding, nothing prevents to replace the control oracle currently used with classic MDP planning methods used in RL. In conclusion, we agree with the Reviewer that the choice of introducing the problem with RL notation is mostly a notational convention.
>
> Ultimately, we want to thank the Reviewer for reporting a typo, which we have corrected, and mentioning several relevant related references of which we were not aware.
>
> **References**
>
> [1] Lucic et al., Are GANs Created Equal? A Large-Scale Study. NeurIPS 2018.
>
> [2] Domingo-Enrich et al., Adjoint Matching: Fine-tuning Flow and Diffusion Generative Models with Memoryless Stochastic Optimal Control. ICLR 2025.
>
> [3] Uehara et al., Feedback Efficient Online Fine-Tuning of Diffusion Models. ICML 2024.
>
> [4] Mutti et al., Challenging common assumptions in convex reinforcement learning. NeurIPS 2022.
>
> [5] Zhang et al., Variational policy gradient method for reinforcement learning with general utilities. NeurIPS 2020.
>
> [6] Karimi et al., Sinkhorn flow as mirror flow: A continuous-time framework for generalizing the sinkhorn algorithm, AISTATS 2024.
>
> [7] Borkar et al., The ODE method for convergence of stochastic approximation and reinforcement learning. SIAM Journal on Control and Optimization 2000.

---

### Decision · Program_Chairs · 2025-05-01

**Decision:**

Accept (poster)

**Comment:**

Given a pre-trained diffusion model, this paper proposes a method to sample from the maximum entropy distribution supported on the support of the diffusion model. Reviewers had several criticisms of the paper which were mostly addressed during the rebuttal, and I thus recommend acceptance.

To the authors: If the paper is accepted, I ask you to please follow the suggestion of reviewer WBXm and include the Vendi score as an evaluation metric. In your rebuttal you mentioned this was difficult because choosing the kernel is not trivial, and that it is unclear if this score correlates with entropy. As for the kernel choice, you can just use the default one in the public implementation. As for the potential lack of correlation with entropy, although you are technically correct, it would be quite surprising to observe it in practice and this would warrant additional discussion.